# Interleukin-1 receptor-induced PGE$_2$ production controls acetylcholine-mediated cardiac dysfunction and mortality during scorpion envenomation

Mouzarllem B. Reis [1,2,11], Fernanda L. Rodrigues[3,4], Natalia Lautherbach[3,5], Alexandre Kanashiro [6], Carlos A. Sorgi [1,12], Alyne F. G. Meirelles[1], Carlos A. A. Silva[3], Karina F. Zoccal[7], Camila O. S. Souza [1,2], Simone G. Ramos[8], Alessandra K. Matsuno [9], Lenaldo B. Rocha [10], Helio C. Salgado[3], Luiz C. C. Navegantes[3], Ísis C. Kettelhut[3,5], Palmira Cupo [9], Luiz G. Gardinassi [1,2,13] & Lúcia H. Faccioli [1,2 ✉]

Scorpion envenomation is a leading cause of morbidity and mortality among accidents caused by venomous animals. Major clinical manifestations that precede death after scorpion envenomation include heart failure and pulmonary edema. Here, we demonstrate that cardiac dysfunction and fatal outcomes caused by lethal scorpion envenomation in mice are mediated by a neuro-immune interaction linking IL-1 receptor signaling, prostaglandin E$_2$, and acetylcholine release. IL-1R deficiency, the treatment with a high dose of dexamethasone or blockage of parasympathetic signaling using atropine or vagotomy, abolished heart failure and mortality of envenomed mice. Therefore, we propose the use of dexamethasone administration very early after envenomation, even before antiserum, to inhibit the production of inflammatory mediators and acetylcholine release, and to reduce the risk of death.

[1] Departamento de Análises Clínicas, Toxicológicas e Bromatológicas, Faculdade de Ciências Farmacêuticas de Ribeirão Preto, Universidade de São Paulo, São Paulo, Brazil. [2] Programa de Pós-Graduação em Imunologia Básica e Aplicada, Faculdade de Medicina de Ribeirão Preto, São Paulo, Brazil. [3] Departamento de Fisiologia, Faculdade de Medicina de Ribeirão Preto, Universidade de São Paulo, São Paulo, Brazil. [4] Faculdade de Medicina, Campus de Três Lagoas/CPTL, Universidade Federal de Mato Grosso do Sul, Mato Grosso do Sul, Brazil. [5] Departamento de Bioquímica e Imunologia, Faculdade de Medicina de Ribeirão Preto, Universidade de São Paulo, São Paulo, Brazil. [6] Departamento de Neurociências e Ciências do Comportamento, Faculdade de Medicina de Ribeirão Preto, Universidade de São Paulo, São Paulo, Brazil. [7] Centro Universitário Barão de Mauá, Ribeirão Preto, São Paulo, Brazil. [8] Departamento de Patologia e Medicina Legal, Faculdade de Medicina de Ribeirão Preto, São Paulo, Brazil. [9] Departamento de Puericultura e Pediatria, Faculdade de Medicina de Ribeirão Preto, São Paulo, Brazil. [10] Instituto de Ciências Biológicas e Naturais, Universidade Federal do Triângulo Mineiro, Minas Gerais, Brazil. [11] Present address: Centro Universitário Barão de Mauá, Ribeirão Preto, São Paulo, Brazil. [12] Present address: Departamento de Quimica, Faculdade de Filosofia. Ciências e Letras de Ribeirão Preto, Universidade de São Paulo, São Paulo, Brazil. [13] Present address: Instituto de Patologia Tropical e Saúde Pública, Universidade Federal de Goiás, Goiânia, Brazil. ✉email: faccioli@fcfrp.usp.br

Severe scorpion envenomation induces pulmonary edema and cardiac dysfunction that can progress to fatal outcomes, especially in children and the elderly[1]. *Tityus serrulatus* scorpion venom (TsV) is recognized by pattern-recognition receptors (PRRs), which induce the production of cytokines and lipid mediators, such as eicosanoids[2,3]. TsV recognition by macrophages results in NLRP3-inflammasome activation and interleukin-1β (IL-1β) release, promoting lung edema and mortality[4]. Inhibition of prostaglandin $E_2$ (PGE$_2$) production or administration of leukotriene $B_4$ (LTB$_4$) suppresses IL-1β release, abrogating both TsV-induced lung edema and mortality[4]. Unfortunately, the development of new treatments has limitations, mainly due to a lack of information about the mechanisms involved in the etiology of clinical manifestations. This includes cardiac dysfunction, which is considered one of the main pathophysiological events associated with the high index of scorpion envenomation-induced mortality.

Currently, it is known that toxins present in TsV activate ion channels that induce a neuroexcitatory syndrome known as a neurotransmitter storm[1,5], an excessive response associated with hyperactivation of the autonomic nervous system, which leads to TsV-induced cardiogenic shock and other systemic manifestations. The autonomic nervous system is divided into parasympathetic and sympathetic systems that perform opposite functions in the maintenance of homeostasis through the release of acetylcholine (ACh) and catecholamines[6,7]. The excessive release of ACh is responsible for bradycardia, cardiac arrhythmias, elevated nasal, salivary, lachrymal, pancreatic, and bronchial secretions, along with piloerection and muscle spasms, while exaggerated catecholamine production mediates cardiac arrhythmias, tachycardia, arterial hypertension, shock, hyperglycemia, and leukocytosis[1,8]. Depending on the intensity of these manifestations, envenomed mammals can progress to severe arterial hypotension and, subsequently, death[1,4,8]. Although the contribution of these neurotransmitters to cardiogenic shock and death has been recognized during severe scorpion envenomation, the molecular mechanisms underlying their transmission remain unclear. Recent progress in interdisciplinary research has described the relevance of (i) the immune system in the physiological control of the cardiovascular system and (ii) neural signals in the regulation of cardiovascular functions, as well as in the heart immune response[9–11]. With these studies in mind, we hypothesized that TsV-induced cardiac dysfunction and subsequent mortality occur through neuroimmune interactions via a specific brain–heart neural circuit. By combining genetic, surgical, and pharmacological approaches, we set out to investigate the neurotransmitter storm mechanistically by exploring the hierarchy of neural and molecular events connecting autonomic with the heart innate immune response.

Here, we show that scorpion envenomation induces autonomic hyperactivation and heart inflammation in mice. Cardiac fibroblasts respond to TsV and produce PGE$_2$ and IL-1β in vitro. Furthermore, IL-1R signaling is required for sustained PGE$_2$ production, amplification of IL-1β release, cardiac dysfunction, and mortality in vivo. Indeed, PGE$_2$ signaling via EP2/4 receptors mediates excessive ACh release, which in turn causes the heart failure of envenomed mice. Of importance, we introduce a novel pathway that is targeted by dexamethasone, but not by anti-scorpion venom serum. Taken together, our study indicates that suppression of the inflammatory response can be an effective supportive therapy to prevent cardiac dysfunction and mortality caused by scorpion envenomation.

## Results

### TsV provokes autonomic dysfunction and heart inflammation.
Lethally envenomed C57Bl/6 (WT) mice (TsV, 180 µg kg$^{-1}$) exhibited sweating, ocular secretion, nasal secretion, lethargy, and convulsion, referred to here as systemic autonomic manifestations in order to create a clinical score and evaluate the severity of envenomation (Fig. 1a). We also detected hyperglycemia (Fig. 1b), a clinical sign that is also observed in scorpion-envenomed patients[8]. Next, to investigate the impact of venom administration on the autonomic systems, we measured the amount of sympathetic and parasympathetic neurotransmitters in the blood, specifically catecholamines (adrenaline and noradrenaline) and ACh, respectively (Fig. 1c–e). We detected a dramatic increase in the concentration of adrenaline and ACh compared to PBS-inoculated mice (control) (Fig. 1c, e). Surprisingly, although studies have suggested that these changes are caused by excessive activation of both the sympathetic and parasympathetic systems[1,8], no studies have identified the mechanisms involved in these scorpion-induced autonomic disturbances, particularly those involved with the cardiovascular system. In lethally envenomed mice, within 60 min, we observed cardiac disorders with diminished mean arterial pressure and heart rates (Fig. 1f, g), which rapidly culminated in death in 50–100% of cases. We also detected peripheral blood leukocytosis (a hematologic manifestation that is also provoked by sympathetic hyperactivation), mainly due to neutrophilia and also due to a reduction in the number of mononuclear cells (Fig. 1h, j). Given that one of the leading causes of death by scorpion envenomation is cardiovascular dysfunction, we sought to determine the effect of scorpion venom on the induction of an inflammatory response in the heart. Just as we observed in the blood, flow-cytometry analysis of leukocytes in the heart demonstrated an increase in the neutrophil influx and a decrease in macrophages (Fig. 1k, l), which were accompanied by the upregulation of genes involved in innate immune responses and eicosanoid metabolism (Fig. 1m and Supplementary Table 1). Moreover, TsV increased NLRP3, Asc, and Casp1/11−dependent IL-1β release (Fig. 1n), as well as PGE$_2$ production in the heart (Fig. 1o) and PGE$_2$ in the serum (Fig. 1p). These data confirm that lethally envenomed mice present clinical autonomic manifestations similar to those observed in humans[1,8], and that TsV induces a pro-inflammatory response in the heart, an organ richly innervated by both branches of the autonomic system.

### Cardiac fibroblasts mediate the inflammatory response to TsV.
Given the pro-inflammatory profile observed in the hearts of TsV-envenomed animals, next, we wished to determine which cardiac cell(s) contribute to the inflammatory process. Upon stimulating primary cultures of either cardiomyocytes (CMs) or cardiac fibroblasts (CFs) with TsV for 24 h, we found that only CFs released IL-1β (Fig. 2a), while both cell types produced PGE$_2$ (Fig. 2b). Moreover, TsV-stimulated CFs exhibited higher expression of inflammatory genes and increased cell surface expression of CD14 and TLR4, but not TLR2 (Supplementary Table 2 and Fig. 2c, respectively). It is known that at least two signals are required to activate the NLRP3 inflammasome and to boost IL-1β maturation and release by macrophages[12]. Following TsV exposure, these signals include PRR signaling, potassium (K$^+$) efflux, PGE$_2$ release, production of 3′, 5′-cyclic adenosine monophosphate (cAMP), protein kinase A (PKA) activation, and enhanced nuclear factor-κB (NF-κB) activity[2,4]. Interestingly, TsV-stimulated CFs also require K$^+$ efflux and PGE$_2$-EP2/cAMP/PKA signaling to produce IL-1β, which was impaired under culture conditions with elevated concentrations

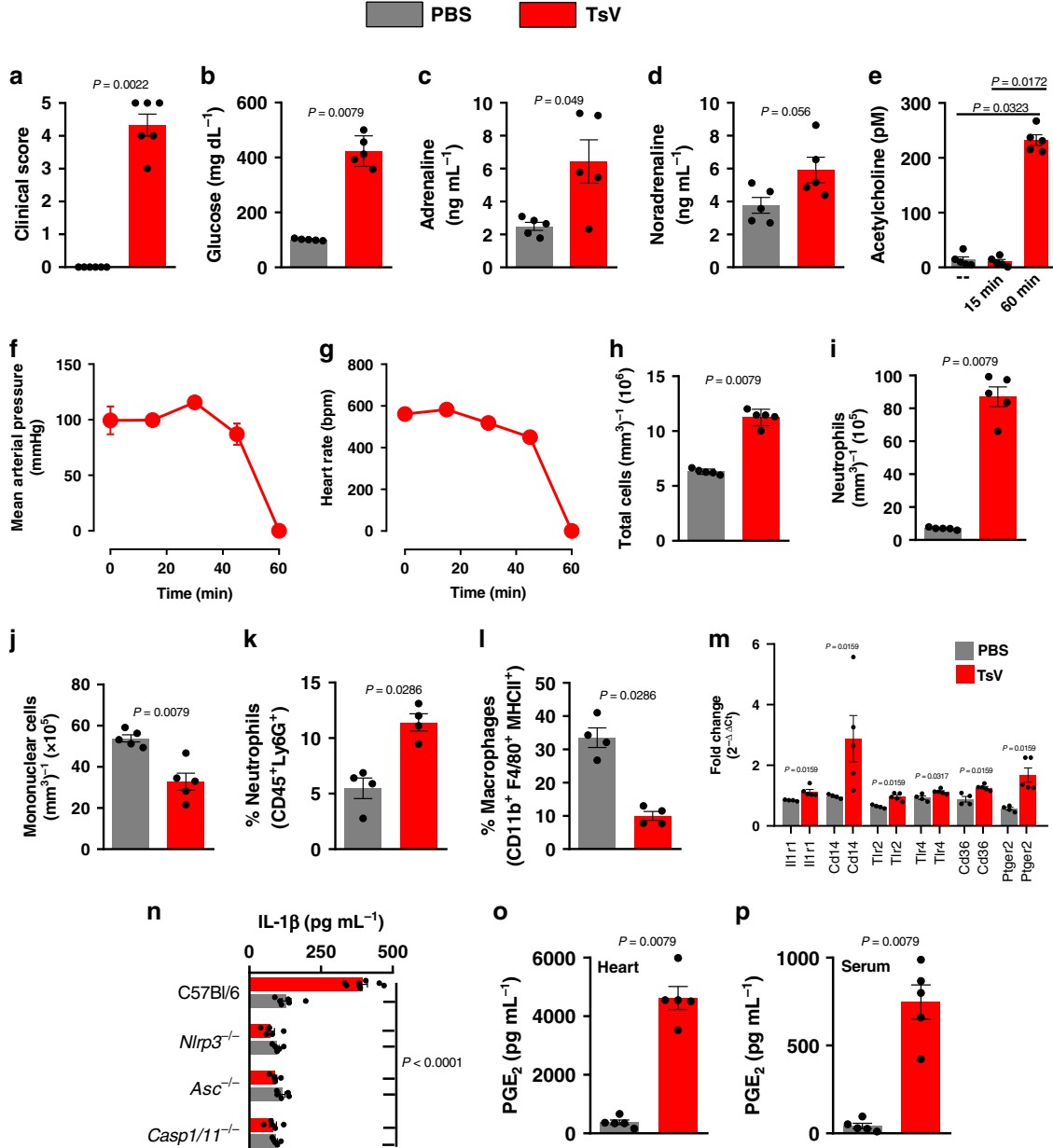

**Fig. 1 TsV induces systemic manifestations and heart inflammation.** C57Bl/6 (or indicated strain) mice were challenged with PBS or with a lethal dose of TsV and observed for 60 min, or the time indicated. At the time specified, blood was collected for the measurements. **a** Intense sweating, ocular secretion, nasal secretion, convulsion, and lethargy were recorded in order to compute a clinical score. **b** Glucose concentration in plasma, 60 min after the stimuli (**a**, **b**, $n = 5$–6, one representative of three experiments). **c** Adrenaline and **d** noradrenaline in plasma were quantified by HPLC, 60 min after the stimuli. **e** Acetylcholine in plasma was quantified using a commercial kit 15 or 60 min after the stimuli (**c**–**e**, $n = 5$, one representative of three experiments). **f** Mean arterial pressure and (**g**) heart rates were monitored for 80 min ($n = 3$, one experiment). **h** Total leukocyte, **i** neutrophil, and **j** mononuclear cell counts in peripheral blood collected 60 min after the stimuli (**h**–**j**, $n = 5$, one experiment). **k** Percentage of neutrophils (gated as $CD45^+ Ly6G^+$) and **l** macrophages (gated as $CD45^+ F4/80^+ MHCII^+$) assessed by flow cytometry of cell suspensions from a heart harvested 60 min after stimuli (**k**, **l**, $n = 4$, one experiment). **m** Heart gene expression ($n = 4$ for PBS and $n = 5$ for TsV from one experiment). **n** The concentration of IL-1β in heart homogenates of C57Bl/6, $Casp1/11^{-/-}$, $Nlrp3^{-/-}$, and $Asc^{-/-}$ mice ($n = 5$–8, one experiment). $PGE_2$ concentration in (**o**) heart homogenates and in (**p**) serum ($n = 4$–5, representative of two experiments). In all experiments, mice were inoculated with a lethal dose of TsV (180 µg kg$^{-1}$ i.p./300 µl, red bars/lines) or the same volume of PBS (300 µl, i.p., gray bars). Data are expressed as means ± SEM. Unpaired Student's $t$ test with Mann–Whitney were applied to panels **a**–**d**, **h**–**m**, **o**, **p**, and one-way ANOVA followed by Bonferroni's multicomparison test for panels **e**, **m**. Exact $P$ values are expressed in figures, with $P < 0.05$ was considered statistically significant.

of extracellular $K^+$ (but not NaCl) (Fig. 2d), an antagonist of prostaglandin receptors (compound AH6809) (Fig. 2e), or a PKA inhibitor (compound H89) (Fig. 2g). Moreover, in TsV-stimulated CFs, cAMP levels increased after the addition of exogenous $PGE_2$ (Fig. 2f). We also observed the upregulated expression of total and phosphorylated forms of PKA in heart

lysates of envenomed mice (Fig. 2h). These data suggest that TsV causes CFs to release IL-1β via NLRP3-inflammasome activation, which can be enhanced by the $PGE_2$/cAMP/PKA/NF-κB pathway, demonstrating that, aside from systemic inflammation, TsV induces heart inflammation by stimulating CFs to produce IL-1β and $PGE_2$.

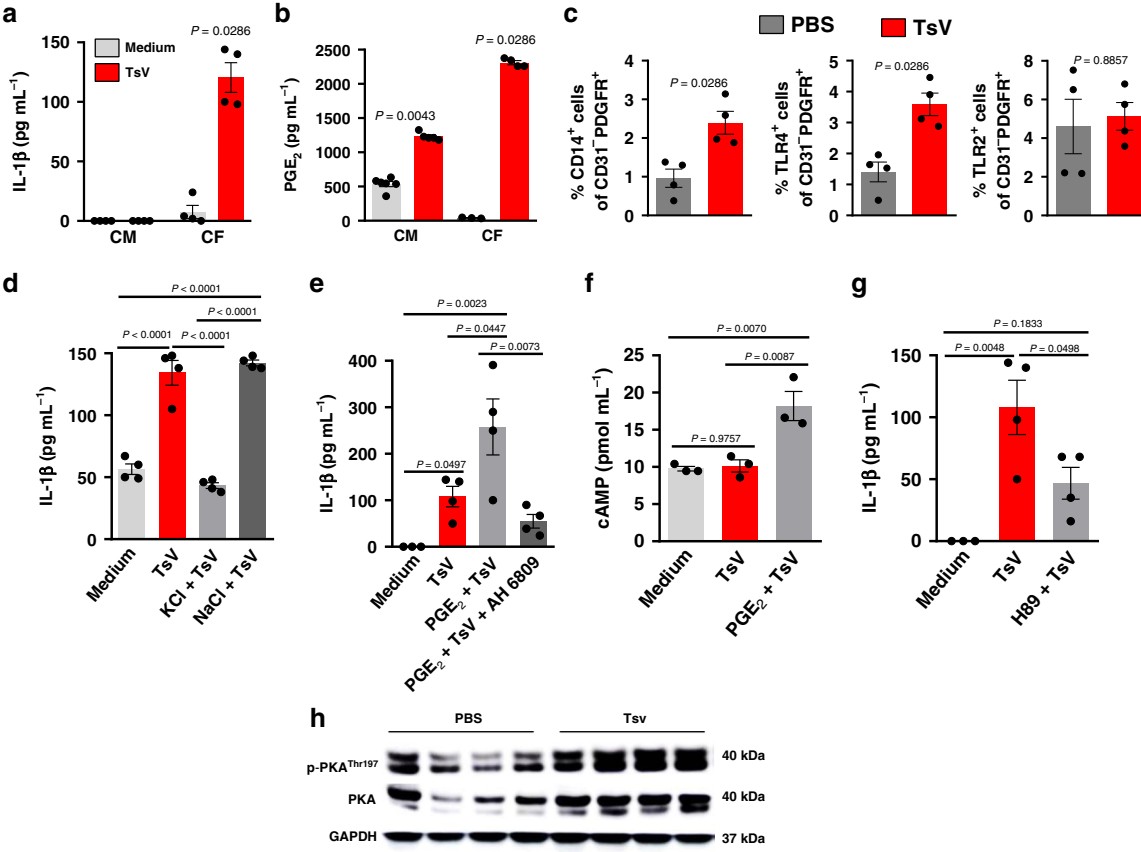

**Fig. 2 Cardiac fibroblasts release PGE2 and IL-1β upon TsV stimulation. a** IL-1β and **b** PGE$_2$ concentration in supernatants of primary cultures of cardiac fibroblasts (CFs) and cardiomyocytes (CMs) incubated with medium or TsV (**a**, **b**, $n = 4$–5 wells/per condition, one experiment). **c** Percentage of CFs (gated as CD31$^-$PDGFR$^+$) expressing CD14$^+$, TLR4$^+$, or TLR2$^+$ as assessed by flow cytometry of heart cell suspensions from mice inoculated with PBS or TsV ($n = 4$/group one experiment). **d** IL-1β concentration in the supernatants of CFs incubated with medium or TsV with or without KCl (50 mM) or NaCl (50 mM). **e** IL-1β concentration in supernatants of CFs pre-treated or not with PGE$_2$ 10 μM 10 min before TsV, in the presence or absence of EP2 antagonist (AH6908, 1 μM, 30 min before PGE$_2$), followed by stimulation with TsV (**e**, **f**, $n = 4$ wells/per condition, one experiment). **f** cAMP in primary CFs, pre-treated or not with PGE$_2$ (10 μM) 10 min before TsV stimulation ($n = 3$ wells/per conditions, one experiment). **g** IL-1β concentration in supernatants of CFs, pre-treated or not with PKA inhibitor (H89, 25 μM) 120 min before stimulation with TsV ($n = 4$ wells/per conditions, one experiment). For in vitro experiments, CF or CM cultures (**a–c**, **e–h**) were incubated with medium, as a control, or TsV (50 μg ml$^{-1}$) for 24 h, except for cAMP quantification (5 min). Data are expressed as means ± SEM. **h** Western blotting analyses of total and phosphorylated PKA in heart homogenates from mice inoculated with PBS or TsV ($n = 4$ mice/group, one representative of two experiments). For in vivo experiments (**d**, **i**), mice were inoculated with a lethal dose of TsV (180 μg kg$^{-1}$ i.p./300 μl) or PBS (300 μl, i.p.), as a control, and euthanized after 60 min. C57Bl/6 mice were used for in vitro and in vivo experiments. Data are expressed as means ± SEM. Unpaired Student's $t$ test with Mann–Whitney were applied to panel **c**, and one-way ANOVA followed by Bonferroni's multi-comparison test for panels **a**, **b**, **d–g**. Exact $P$ values are expressed in figures, while $P < 0.05$ was considered statistically significant.

**Impact of the cardiac inflammation induced by envenomation.** To determine the impact of local and systemic inflammation on heart function, we used mice deficient for the IL-1 receptor (IL-1R) or suppressed the inflammatory response with a high dose of corticosteroid. Deficiency of IL-1R ($Il1r1^{-/-}$) or treatment of WT-envenomed animals with dexamethasone (DEX) abolished the previously observed alterations in mean arterial pressure and heart rate (Fig. 3a, b). The echocardiographic analysis showed that only untreated-WT-envenomed mice presented cardiac dysfunction, as estimated by reductions in the stroke volume (SV; the volume of blood pumped out of the left ventricle of the heart during each systolic cardiac contraction), ejection fraction (EF; measurement, expressed as a percentage, of how much blood the left ventricle pumps out with each contraction) and cardiac output (CO; the blood volume the heart pumps through the systemic circulation over a period measured in liters per minute) (Fig. 3c, d). Interestingly, WT-envenomed animals treated with DEX or $Il1r1^{-/-}$-envenomed mice were resistant to fatal outcomes (Fig. 3e, j, respectively) and exhibited reduced clinical

scores (Fig. 3f, k, respectively). Resistance to TsV envenomation correlated with a significant reduction in the concentration of IL-1β in the heart (Fig. 3g, l, respectively), as well as in the concentration of PGE$_2$ in the heart (Fig. 3h, m, respectively) and the serum (Fig. 3i, n, respectively). Collectively, these data indicate the pivotal role of inflammation in cardiac dysfunction and mortality during severe scorpion envenomation.

The above-mentioned results revealed the crucial role of inflammation in heart failure induced by scorpion venom. Therefore, we hypothesized that rapid inhibition of venom-induced inflammation might be a useful therapy to treat scorpion envenomation, in addition to antiserum administration. To explore the potential use of an anti-inflammatory as a supportive therapy for scorpion envenomation, a proof-of-concept experiment was conducted. Lethally envenomed animals showed a decrease in critical cardiac parameters, such as ejection fraction, stroke volume, and cardiac output (Fig. 4a), and, as expected, DEX treatment 30 min after TsV inoculation prevents these venom-induced cardiac dysfunctions (Fig. 4b). Interestingly, the

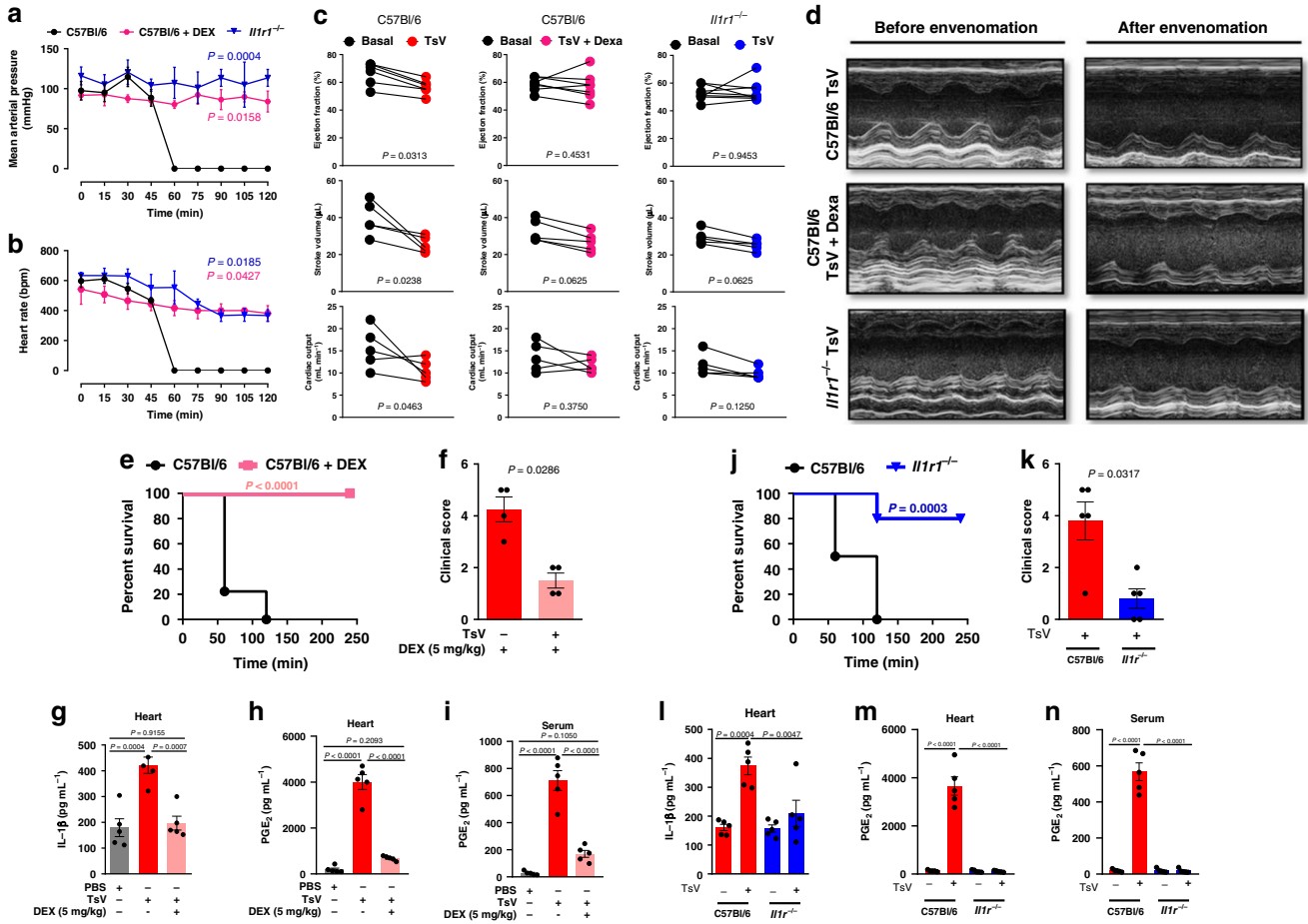

**Fig. 3 TsV-induced IL-1β/IL-1R signaling promotes cardiac dysfunction. a** Arterial pressure and **b** heart rate were recorded in $Il1r1^{-/-}$ or C57Bl/6 mice, treated or not with DEX, and challenged i.p. with a lethal dose of TsV ($n = 4$-5, one experiment). Envenomed mice had their echocardiography examination performed for 15 min, starting 45 min after envenomation and ending 60 min after TsV. **c** Ejection fraction (EF), stroke volume (SV), and cardiac output (CO) ($n = 5$-6, one experiment). **d** Representative M-mode image of echocardiographic examination of the left ventricle before and 45 min after inoculation with TsV in $Il1r1^{-/-}$ or C57Bl/6 mice, treated or not with DEX. **e** Survival curves ($n = 10$, one representative of two experiments), **f** clinical score ($n = 5$, one representative of two experiments), **g** IL-1β, and **h** PGE$_2$ concentration in heart homogenates ($n = 5$, one experiment), and **i** PGE$_2$ in the serum of C57Bl/6 mice inoculated with PBS or TsV and treated or not with DEX ($n = 5$, one representative of two experiments). **j** Survival curves ($n = 10$, representative of two experiments), **k** clinical score ($n = 5$, representative of two experiments), **l** IL-1β, and **m** PGE$_2$ in heart homogenates ($n = 5$, one experiment), and **n** PGE$_2$ in the serum of $Il1r1^{-/-}$ mice inoculated with PBS or TsV ($n = 5$, representative of two experiments). In all experiments, mice were inoculated with a lethal dose of TsV (180 μg kg$^{-1}$ i.p./300 μl) or the same volume of PBS. When indicated, DEX (5 mg kg$^{-1}$ i.p.) was given therapeutically 15 min after the venom. Data are expressed as means ± SEM. Unpaired Student's $t$ test with Mann–Whitney were applied to panels **f**, **k**. Paired $t$ test was applied to panel **c**. One-way ANOVA followed by Bonferroni's multicomparison test for panels **a**, **b**, **e**, **j**, **g**–**i**, **l**–**n**. For survival (**e**, **j**), a log-rank test was performed. Exact $P$ values are expressed in figures, while $P < 0.05$ was considered statistically significant.

administration of specific anti-scorpion venom serum modified the drop in ejection fraction, but not the stroke volume or the cardiac output alterations induced by the venom (Fig. 4c). However, DEX treatment 15 min after the venom inoculation and 15 min before the subsequent administration of anti-scorpion venom serum (given 30 min after TsV) prevented heart failure (Fig. 4d). The echocardiographic images reflect the effect of the treatments on mouse heart function (Fig. 4e), whereby treatment with DEX preserved all the features and characteristics of a standard cardiac chamber. Taken together, these results suggest that DEX has the potential to be used as supportive therapy to prevent the occurrence of potentially lethal cardiac changes.

**TsV induces heart failure via PGE$_2$-dependent ACh release.** As revealed before in clinical studies (and reproduced experimentally in this study; Fig. 1a–e), scorpion envenomation induces autonomic hyperactivation[1,5,13]. In our research, we also found increased concentrations of catecholamines (adrenaline) and ACh

in the plasma of lethally envenomed WT mice (Fig. 1c, e); however, the molecular mechanisms through which these neurotransmitters regulate cardiac dysfunction, clinical score, and mortality are still unknown. We verified that antagonizing beta-adrenoceptors with propranolol improved neither animal survival nor the clinical score or high cardiac IL-1β production (Fig. 5a–c). Interestingly, in contrast to treatment with propranolol, the administration of atropine, a nonselective muscarinic antagonist, rescued the mice from mortality (Fig. 5d) and reduced the clinical score (Fig. 5e) without interfering with IL-1β production in the heart (Fig. 5f). Additionally, atropine administration prevented all the observed and evaluated cardiac alterations, reflected by preserved EF, SV, and CO, as observed via echocardiographic analysis (Fig. 5g, h).

The vagus nerve is the main parasympathetic nerve and the most important neural source of peripheral ACh[14], which exerts its cardiac functions mainly by muscarinic receptor activation. Given that muscarinic receptors participate in scorpion venom-

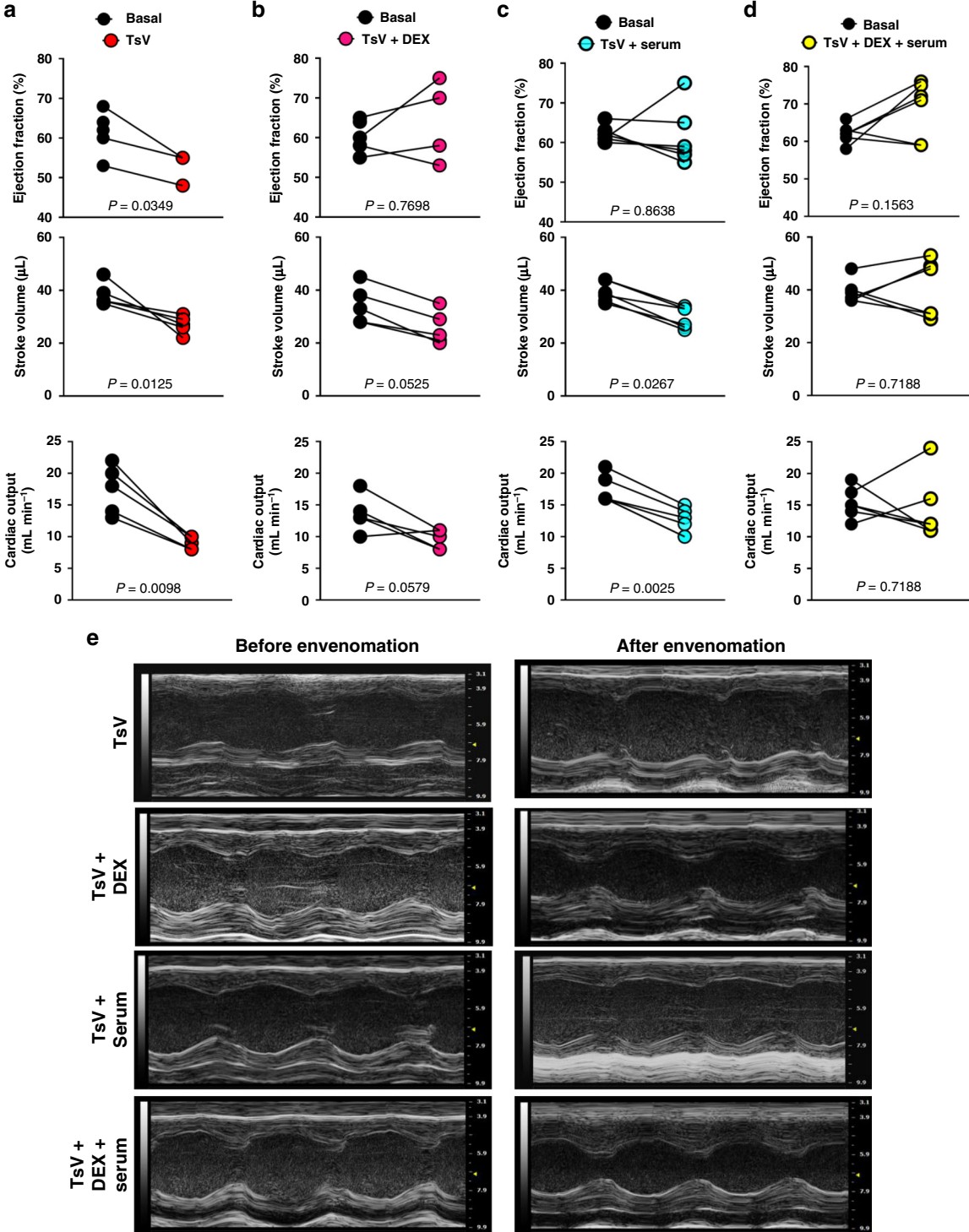

**Fig. 4 Efficacy of therapy with dexamethasone and/or antiscorpionic serum.** Ejection fraction (EF), stroke volume (SV), and cardiac output (CO) of C57Bl/6 mice were submitted to echocardiography before (basal) and 45 min after inoculation with the venom. **a** TsV only; **b** TsV plus DEX (5 mg kg$^{-1}$ i.p.)/15 min after the venom; **c** TsV plus antiscorpionic serum (30 µl/1 mg/ml/animal)/30 min after the venom; or **d** TsV plus DEX (5 mg kg$^{-1}$ i.p.)/15 min after the venom followed by antiscorpionic serum (30 µl/1 mg/ml/animal)/15 min after DEX ($n = 5$–6, one experiment). **e** Representative M-mode image of echocardiographs obtained from the above-mentioned groups. Data are expressed as means ± SEM. Paired $t$ test was applied to evaluate statistical significance. Differences were considered significant if $P < 0.05$.

induced autonomic dysfunctions, we next investigated the involvement of parasympathetic signaling after TsV envenomation. It has already been described that right cervical vagotomy decreases peripheral ACh concentration[15]; thus, we took advantage of this knowledge to define the relationship between

the vagus nerve and TsV envenomation. Mice were vagotomized or sham-operated 5 days before lethal TsV envenomation. We observed that right cervical vagotomy protected the mice from envenomation, preserving the cardiac functions (EF and CO), as demonstrated by echocardiographic analysis (Fig. 5g, h).

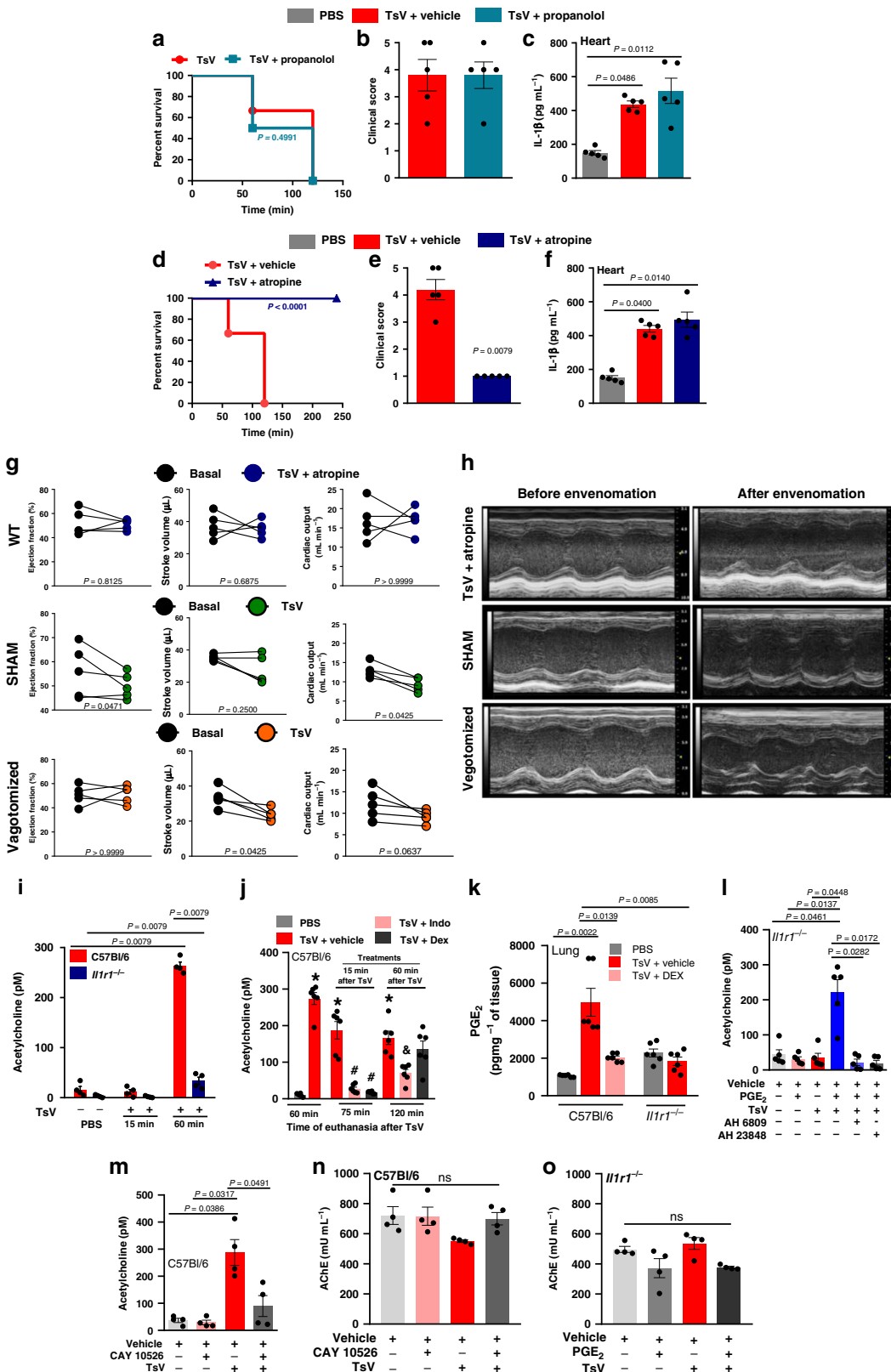

Together, these results suggest that both pharmacological (atropine) and surgical (vagotomy) approaches can block the inflammation-regulated parasympathetic signaling that controls severe cardiac dysfunction.

Since IL-1R deficiency and DEX treatment prevent $PGE_2$ production, cardiac dysfunction, and mortality, and with the knowledge that $PGE_2$ stimulates neuronal ACh release[16,17], we next investigated its association with ACh release in scorpion envenomation. Interestingly, the absence of IL-1R-signaling and treatment with DEX or indomethacin (Indo, a COX1/2 inhibitor) both abrogated TsV-induced systemic ACh release (Fig. 5i, j) and $PGE_2$ production in the lungs (Fig. 5k). Because indomethacin

**Fig. 5 PGE2-EP2/EP4 signaling controls ACh release and cardiac dysfunction.** In one set of experiments, C57Bl/6 mice were pre-treated with (**a–c**) vehicle or propranolol (4 mg kg$^{-1}$ i.p., 60 min before), or (**d–f**) vehicle or atropine (2 mg kg$^{-1}$ i.p., 60 min before), and then inoculated with TsV and observed for 240 min for (**a, d**) survival curves ($n = 10$), or for 60 min for (**b, e**) clinical score. **c, f** IL-1β concentration in heart homogenates. **b**; **c**; **e**, **f**, $n = 4$–5 one experiment. **g** Naive C57Bl/6 mice pre-treated with atropine (2 mg kg$^{-1}$ i.p., upper panel) were inoculated 60 min after with a lethal dose of TsV ($n = 5$), and a group of sham-operated (middle panel) or vagotomized animals (bottom panel) ($n = 5$) were inoculated with a lethal dose of TsV for evaluation of ejection fraction (EF), stroke volume (SV), and cardiac output (CO) ($n = 5$, one experiment). **h** Representative M-mode image of echocardiographic examination of the left ventricle of one mouse from each group described in (**g**). In a different set of experiments, **i** C57Bl/6 or *Il1r1*$^{-/-}$ mice were inoculated with PBS or TsV for quantification of acetylcholine (ACh) in plasma at 15 and 60 min after envenomation ($n = 5$, representative of two experiments). **j** C57Bl/6 were inoculated with TsV and 15 or 60 min after were treated with vehicle, indomethacin (2 mg kg$^{-1}$ i.p.), or dexamethasone (5 mg kg$^{-1}$ i.p.) for quantification of ACh in the plasma 75 and 120 min after the venom inoculation. ($n = 6$, from two experiments with three mice each time). *$P < 0.0001$, PBS vs TsV + vehicle; #$P < 0.05$, TsV + DEX or TsV + Indo vs TsV + vehicle; &$P < 0.05$, TsV + Indo vs TsV + vehicle. **k** *Il1r1*$^{-/-}$ or C57Bl/6 mice were inoculated with TsV and treated or not with DEX (5 mg kg$^{-1}$ i.p.) for quantification of pulmonary PGE2 ($n = 6$ for C57Bl/6, and $n = 6$ for *Il1r1*$^{-/-}$ mice). **l** ACh was quantified in the plasma of *Il1r1*$^{-/-}$ mice inoculated i.p. with vehicle or TsV and treated or not with PGE2 (1 mg kg$^{-1}$ i.p., 15 min after TsV inoculation), in the presence or not of EP2 (AH6809, 5 mg kg$^{-1}$, 24 h, and 12 h before TsV) or EP4 (AH23848 5 mg kg$^{-1}$ 24 h and 12 h before TsV) antagonists ($n = 4$–5, one experiment). **m** ACh in the plasma of C57Bl/6 mice pre-treated with mPGES-1 inhibitor (compound CAY10526, 5 mg kg$^{-1}$, 24 h, and 60 min before venom) and inoculated with TsV ($n = 4$, one experiment). **n** Acetylcholinesterase activity in the serum of C57Bl/6 mice pre-treated with mPGES-1 inhibitor (compound CAY10526, 5 mg kg$^{-1}$ i.p.) or vehicle 24 h and 60 min before inoculation with TsV. **o** Acetylcholinesterase activity in the serum of *Il1r1*$^{-/-}$ mice pre-treated with PGE2 (1 mg kg$^{-1}$ i.p.) or vehicle 24 h and 15 min before inoculation with TsV. ($n = 5$ in each group, one experiment). In all experiments, mice were inoculated with a lethal dose of TsV (180 μg kg$^{-1}$ i.p./300 μl) or PBS i.p. (300 μl), or vehicle i.p. (300 μl) when indicated. The data are expressed as means ± SEM. Differences were considered significant if $P < 0.05$ according to Student's *t* test (**b, c, e, f**). One-way ANOVA followed by Bonferroni's multicomparison test (**i–o**). Paired *t* test (**g**) or the log-rank test for survival (**a, d**).

treatment of envenomed WT mice inhibited ACh release (Fig. 5j) and *Il1r1*$^{-/-}$ animals produced lower concentrations of PGE2 after TsV envenomation (Zoccal et al.[2]) compared to WT mice (Figs. 3m, n and 5k), we hypothesized that IL-1R-induced PGE2 production controls systemic ACh release during scorpion envenomation. In support of this hypothesis, we discovered that *Il1r1*$^{-/-}$ mice only exhibited increased systemic ACh when TsV was inoculated at the same time as exogenous PGE2 (Fig. 5l). Next, we confirmed the functional role of PGE2 in TsV-induced ACh release pharmacologically by antagonizing EP2/EP4 receptors (AH6809 and AH23848, respectively) in *Il1r1*$^{-/-}$-envenomed mice (Fig. 5l) or by selectively inhibiting microsomal PGE synthase-1 (mPGES-1) (CAY10526) in envenomed WT animals (Fig. 5m). To investigate whether PGE2 can inhibit the AChE activity, we measured AChE activity in the serum. We observed no differences in the activity of serum AChE in either WT or *Il1r1*$^{-/-}$ envenomed mice with or without CAY10526 or PGE2 treatment (Fig. 5n, o). Altogether, our results demonstrate that cardiac dysfunction and mortality induced by TsV depend on a neuroimmune pathway involving IL-1R-induced PGE2 production, which in turn stimulates ACh release via EP2/EP4 signaling. The mechanism is summarized in Fig. 6.

## Discussion

The hyperactivation of the autonomic system and excessive inflammation during scorpion envenomation have been previously documented[1,4]; however, the mechanisms involved in venom-induced cardiovascular dysfunction and mortality through manipulation of the autonomic system have never been investigated, particularly from a neuroimmune perspective. In the present study, we established, for the first time, a direct relationship between the inflammatory process, neuronal activation, cardiac dysfunction, and mortality induced by scorpion venom. Also, we demonstrated that although both the parasympathetic/vagal and sympathetic systems are excessively activated after scorpion venom administration, only the blockage of the first prevented the cardiovascular alterations and mortality typically observed in envenomed mice.

Inoculation of TsV induces systemic changes in WT mice, similar to those observed in humans[1,8]. However, the impact of scorpion venom on heart inflammation and its connection with the release of neuro-mediators release has never been deeply

explored. In this investigation, as previously described in macrophages and lungs[2,4,18], we showed that CFs of WT mice recognized the venom and increased the expression of inflammatory genes coding for COX-2 and EP2 receptor, as well CD14 and TLR4 PPR, in addition to the release of PGE2 and IL-1β, through mechanisms regulated by cAMP and PKA. We also confirmed that the NLRP3-inflammasome platform is responsible for IL-1β production in the heart and CFs via a mechanism similar to that first described in macrophages[4]. Previous studies have shown that CFs express PRRs that recognize pathogen-associated molecular patterns (PAMPs), damage-associated molecular patterns, and venom-associated molecular patterns (VAMPs)[2,19–21], and the increase in NLRP3 expression in CFs has been implicated in the size of the infarcted area during ischemia/reperfusion[22], and is crucial for cardiac dysfunction during sepsis[23]. Interestingly, the role of IL-1β in cardiac dysfunction has already been demonstrated in previous studies[20,24–27], given that this cytokine can desensitize cardiomyocytes after adrenergic stimulation and induce reversible contractile dysfunction in the cardiac muscle along with arrhythmias[24]. However, in addition to the mediators produced by the heart cells, we also have to consider the contribution of PGE2 and IL-1β produced both systemically and by the lungs to cardiac dysfunction. We have previously reported that after scorpion envenomation, the main source of PGE2 are resident macrophages in the lungs, but the contribution of other peripheral mononuclear cells cannot be discarded. However, further studies are necessary to better assess this important aspect[4]. Curiously, and contrary to our previous observation in macrophages[4], TsV did not increase cAMP production by cardiac fibroblasts in vitro, unlike the high amount of exogenous PGE2 in vitro (10 μM). Since cardiac fibroblasts are naturally low PGE2 (5 nM, Fig. 2a) and high cAMP (Fig. 2h) producers[28], a high concentration of PGE2 might be necessary to boost additional cAMP production by heart cells in vivo, and this could be a mechanism to prevent excessive activation of cardiac fibroblasts and interstitial fibrosis[28]. On the other hand, high concentrations of PGE2 are apparently necessary to promote ACh release, as observed in *ll1r1*$^{-/-}$ TsV- and PGE2-inoculated mice (35 μM/per animal, Fig. 5l). In WT TsV-envenomed mice, lungs provided a higher concentration of PGE2.

Challenging IL-1R-deficient animals (*ll1r1*$^{-/-}$) or DEX-treated WT mice with a lethal dose of TsV confirmed the involvement of

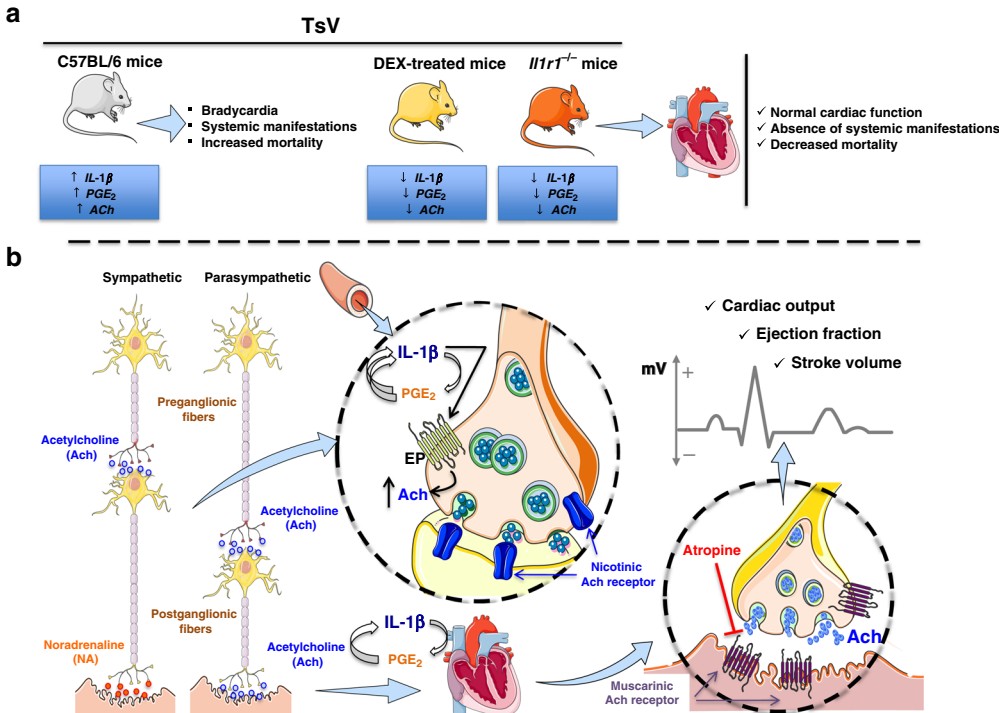

**Fig. 6 Molecular mechanism linking IL-1R-PGE2-ACh axis to cardiac dysfunction.** A lethal dose (180 μg kg$^{-1}$) of scorpion venom (TsV) induces PGE$_2$, IL-1β, and ACh, resulting in cardiac dysfunctions and death. PGE$_2$ via the EP2/4 receptor induces ACh release through either pre-ganglionic fibers of the sympathetic system or pre- and post-ganglionic fibers of the parasympathetic system, which cause decreases in the heart ejection fraction, stroke volume, and cardiac output. Additionally, PGE$_2$-EP2/EP4 amplifies IL-1β production, which in turn, binds to the IL-1R and potentiates PGE$_2$ and ACh. IL-1R (*Il1r1*$^{-/-}$) deficiency or a high dose of DEX blocks PGE$_2$, IL-1β, and ACh release and protects mice from cardiac dysfunction and mortality. Antagonism of muscarinic receptors by atropine and vagotomy prevent the cardiac alterations typically observed during severe scorpion envenomation.

the inflammatory mediators PGE$_2$ and IL-1β in the systemic manifestation, cardiac dysfunction, and mortality (Fig. 3). Because scorpion venom induces the release of catecholamines and ACh[1], and knowing that the autonomic system controls the cardiovascular, gastrointestinal, and urogenital systems and also regulates the immune responses[7,29], we next investigated the contribution of these neurotransmitters to cardiac dysfunctions, clinical score, and mortality. Pharmacological approaches and vagotomy confirmed the relevance of ACh for systemic and cardiac manifestations of severe scorpion envenomation. Additionally, in this study, we revealed that the sequential activation of parasympathetic (vagal) signaling and muscarinic receptors, which are the final steps in the IL-1β/IL-1R/PGE$_2$ inflammatory axis in the heart, promotes severe TsV-induced cardiovascular dysfunction. Different studies have demonstrated interactions between PGE$_2$ and ACh synthesis. For example, EP4 activation in a leukemic T-cell line increases ACh production[30]. Also, it is known that the stimulation of the vagus nerve promotes the synthesis of ACh through a mechanism involving T cells, which act as a non-neural source of this neurotransmitter[31]. Similar to our results, a previous study has demonstrated the relationship between the release of this ACh and mPGES-1-dependent PGE$_2$ production in neuroimmune vagal activity[32]. However, findings in the literature are controversial. Some studies confirm that electrical stimulation of the vagus nerve controls exacerbated inflammatory responses[33], such as in rheumatoid arthritis and Crohn's disease[34,35]. Other neuroimmune pathways and bioelectronics approaches involving autonomic signaling have been studied in inflammatory conditions[36–38]. In contrast, other studies have demonstrated that autonomic dysfunctions are involved in the pathophysiology of different immunosuppressed and cardiovascular conditions mediated by neuroimmune circuits. For

example, hyperactivation of the parasympathetic and sympathetic nervous systems after acute central nervous system injury, like stroke and spinal cord injury, leads to an immunosuppressive state favoring infectious complications, such as pneumonia[39–41]. Also, chronic psychosocial stress activates sympathetic activity, promoting the differentiation of hematopoietic stem cells and acceleration of the formation of plaques, such as those found in atherosclerosis[42]. Finally, it is crucial to mention that, although ACh has been initially described as an anti-inflammatory neurotransmitter[33], current data have suggested contradictory conclusions[43]. The data described in this paper are fundamental for the comprehension of human scorpion accidents because they reveal for the first time a neuroimmune response influencing the parasympathetic pathway after scorpion envenomation, which is responsible for cardiac failure. In agreement with our results, a recent study described a neuroimmune cholinergic pathway mediated by sympathetic, but not parasympathetic, activation responsible for immune activation, and hypertension development[9]. It is plausible that hyperactivation of the cholinergic pathways mediates excessive ACh (direct) or catecholamine (indirect) release after inflammatory stimuli, generating opposite hemodynamic effects and therefore promoting severe hypotensive and hypertensive responses, respectively. Studying interactions between the nervous and immune systems has introduced new concepts for organism functioning and has suggested neuroimmune approaches as attractive alternative treatments. In this study, surgical (vagotomy) and pharmacological (atropine) blockage of parasympathetic signals improved TsV-induced cardiac dysfunctions and mortality. Curiously, previous studies have demonstrated that blocking neural signaling protects against infectious and autoimmune diseases[44,45]. Our pharmacological findings highlight the possibility that muscarinic antagonists may

have a positive impact on mortality and several cardiovascular dysfunctions after venom administration. However, although supportive therapy with atropine has already been indicated for severe bradycardia and hypotension, it must be used with caution[1] to avoid unwanted side effects such as dry mouth, nausea, and death. On the other hand, muscarinic antagonists and agonists involved in the vagal activation or cardiac cells could be used selectively to reduce side effects[46]. Selective muscarinic-2 antagonist drugs could be candidates for treating scorpion envenomation since this receptor sub-type exerts parasympathetic activities in the heart rate and atrioventricular nodal conduction. However, although we reported here that the blockage of parasympathetic signals by vagotomy or atropine administration improves the survival of envenomed mice, the precise neuroimmune mechanism by which $PGE_2$-EP2/ EP4 signaling induces ACh release remains to be determined.

Typically, the treatment of scorpion envenomation involves only the administration of antivenom serum, and in some hospitals, a non-immunosuppressive corticosteroid dose is given at the same time as the serum to prevent serum sickness[8]. However, although the administration of antivenom serum is a mandatory treatment to neutralize neurotoxins presents in scorpion venom[8], it is not easily accessible in several clinical ambulatories. Moreover, the antiserum administration to envenomed patients does not control the massive release of inflammatory mediators. Here, as a proof-of-concept, we demonstrated that despite the beneficial impact of the antiserum administration, it addresses only the decrease of ejection fraction (Fig. 4). In contrast, the administration of a high dose of the anti-inflammatory DEX, which inhibits $PGE_2$ and IL-1β (Fig. 3), 30 min after challenging the mice with a lethal dose of scorpion venom, but before administration of the antiserum, prevents cardiac dysfunction. Interestingly, the administration of DEX before the antiserum restores the cardiac function to the same level as non-envenomed animals. DEX is the most effective anti-inflammatory drug used to inhibit $PGE_2$ and IL-1β[3,4], is used worldwide, has a low cost, and is easily found in the most remote regions of the globe.

Therefore, in scorpionism, our findings encourage the early administration of a high dose of DEX to block the production of inflammatory mediators, and consequently, the release of ACh and heart failure, especially in situations in which delayed antiserum treatment increases the risk of death. Translational research beyond murine models is needed to explore the potential strategy of neuroimmunomodulation in the treatment of scorpion envenomation.

## Methods

***Tityus serrulatus* venom (TsV)**. Lyophilized venom obtained from yellow scorpions (*Tityus serrulatus*) used in this study was provided by the Butantan Institute, São Paulo, SP, Brazil. Briefly, the venom was extracted by electric stimulation followed by freeze–drying, and was stored at −20 °C. For all experiments, TsV was diluted in sterile PBS (phosphate-buffered saline) and filtered in a 0.22-μm sterilizing membrane (Millipore, USA). Next, LPS detection test was performed (LAL; QCL-1000, Bio Whittaker, Cambrex Company, USA) according to the manufacturer's instructions. In all TsV samples, LPS was not detected.

**Animals**. For the primary culture of cardiac fibroblasts (CFs) and cardiomyocytes (CMs), newborn C57Bl/6 mice (1–2 days) obtained from the animal facilities of the Faculdade de Medicina de Ribeirão Preto, University of São Paulo (FMRP-USP), were used. For in vivo studies, adult male C57Bl/6 (6–8 weeks) and *Il1r1*$^{−/−}$ animals matched by age and weight in all procedures were obtained from the animal facilities of FMRP-USP. Mice were maintained in a light/dark cycle with free access to food and water, room temperature (25 °C), in 40–60% of humidity. Maintenance and the experiments with mice were conducted in accordance with the Ethical Principles in Animal Research adopted by the National Council for the Control of Animal Experimentation (CONCEA), and were approved by the Animal Care and Use Committee of the Faculdade de Ciências Farmacêuticas de Ribeirão Preto (CEUA-FCFRP) at the Universidade de São Paulo (FCFRP-USP),

Ribeirão Preto, São Paulo, Brazil (Process no. 16.1.1081.60.5) and by the Animal Committee of FMRP-USP (002/2018-1).

**Primary cultures of cardiac cells and in vitro pharmacological treatments**. Two hundred newborn mice were used for a yield of 10 plates of 24-well cells. The hearts from neonatal C57Bl/6 mice (1–2 days) were withdrawn within the laminar flow chamber and digested afterward with collagenase type II (Worthington Biochemical Group, Lakewood, USA) digestion buffer. After five cycles of digestion (digestion-centrifugation-exchange of digestion buffer), the CMs and CFs were separated by Percoll® gradient (GE Healthcare, USA), collected, and incubated in a glucose-enriched medium (25 mM) with newborn calf serum (NBCS; 10%) and the antibiotics penicillin and streptomycin (P/S; 1%), and then kept in incubators for 24 h at 37 °C with 5% $CO_2$. For all experiments, CMs ($4 \times 10^5$ cells/well) or CFs ($8 \times 10^5$ cells/well) were incubated with medium or with 50 μg ml$^{−1}$ of TsV for 24 h at 37 °C in a humidified atmosphere containing 5% $CO_2$. The following treatments were performed in different sets of experiments: prostaglandin $E_2$ ($PGE_2$) (Cayman Chemical Company, USA), 10 μM, 10 min before TsV stimulation; prostaglandin $E_2$ receptor 2 (EP2) antagonist AH6809 (Cayman Chemical Company, USA), 1 μM, 30 min before TsV stimulation; KCl/NaCl (Sigma-Aldrich/Merck, USA), 50 mM plus TsV for 24 h; and protein kinase A (PKA) inhibitor, H89 compound (Sigma-Aldrich/Merck, USA), 25 μM, 2 h before TsV stimulation. In all experiments, except for cAMP measurement (5 min), the supernatants were collected for IL-1β and $PGE_2$ quantification 24 h after TsV stimulation. The adhered cells were used for qRT-PCR. The protocol was adapted from Zoccal et al.[4].

**In vivo experiments and drug treatments**. In all in vivo experiments and under all conditions, C57Bl/6 and *Il1r1*$^{−/−}$ adult mice (22–25 g) were weighed and then inoculated with a lethal dose of TsV, 180 μg kg$^{−1}$ intraperitoneally (i.p.). For the survival curve and echocardiographic analyses, mice were observed for 240 and 120 min, respectively, after the venom inoculation. In the time-response experiments, a group of animals was euthanized 15, 30, and 60 min (the last referred to as the humanitarian endpoint, in which symptoms were assessed for classification of clinical score) after TsV, and the hearts were removed, perfused, and homogenized in sterile water for the quantification of IL-1β and $PGE_2$ (Supplementary Fig. 1a, b). Based on the time course of the production of IL-1β and $PGE_2$ after TsV, 60 min (the peaks of the two mediators) was chosen to perform all of the following experiments in vivo and for collecting hearts, lungs, and blood, except for animals treated with indomethacin (Indo) or dexamethasone (DEX) for ACh quantification (Fig. 2l), for which mice were also euthanized 75 and 120 min after the venom inoculation. Hearts, and in some experiments lungs, were employed for RT-qPCR, western blotting, and histological analysis. For the measurement of blood pressure and heart rate, C57Bl/6 and *Il1r1*$^{−/−}$ animals were anesthetized with a mixture of 2% isoflurane and 10% $O_2$ and kept at a controlled temperature. Surgical procedures were performed under aseptic conditions under the vision of a surgical microscope (DF Vasconcelos model MCM 5, São Paulo). The mice were placed in the supine position and submitted to a cervicotomy. Polyethylene PE-10 catheters (0.61-mm external diameter and 0.28 mm internal, Clay-Adams) were inserted into the left carotid artery to record blood pressure. The arterial catheter was connected to a pressure transducer (DPT-100, Deltran®, Utah Medical Products Inc, USA), and the blood pressure was continuously sampled (4 kHz) using a computer equipped with an analog-digital interface (PowerLab/8SP, ADInstruments, Colorado Springs, CO, USA). The baseline blood pressure was measured for 30 min before TsV injection, and recorded for 120 min after TsV inoculation. The animals were then euthanized using cervical displacement, preceded by anesthesia with ketamine hydrochloride (75 mg kg$^{−1}$) and xylazine hydrochloride (10 mg kg$^{−1}$) administered intraperitoneally. The heart rate was derived from the blood pressure measurement. For the echocardiographic approach, the animals were anesthetized with a mixture of 2% isoflurane and 10% oxygen and kept at a controlled temperature. Doppler echocardiographic examination was performed using the Vevo 2100 System echocardiograph (Visual Sonics, Toronto, ON M4N 3N1, Canada) with a 30 MHz probe. The animals were positioned to obtain the long axis and to record images. Using two-dimensional visualization of the left ventricle in the long axis, an M-mode image was obtained. From the M-mode image, the following parameters were obtained (in mm): diastolic interventricular septum thickness, diastolic left ventricular internal diameter, diastolic left ventricular free wall, systolic interventricular septum thickness, left ventricular internal diameter in systole, and left ventricular free wall thickness in systole. The shortening fraction (%) was calculated from these values, and the ejection fraction, stroke volume, and cardiac output were calculated by the Teichholz method. In all pharmacological treatments, 300 μl of the specific dilutor of each drug (vehicle) or PBS was injected in animals with or without TsV inoculation. The following pharmacological treatments were administered in C57Bl/6 or *Il1r1*$^{−/−}$ mice: Dexamethasone (Aché Pharmaceutical Laboratories, SP, Brazil), 5 mg kg$^{−1}$, i.p., given therapeutically 15 or 60 min after TsV inoculation; mPGES-1 specific inhibitor CAY10526 (Cayman Chemical Company, USA), 1% alcohol in PBS, 5 mg kg$^{−1}$ i.p[47]. 24 h and 60 min before TsV inoculation; $PGE_2$ (Cayman Chemical Company, USA), (PBS + 0.01% of ethanol, 1 mg kg$^{−1}$ i.p[48]. Twenty-four hours and 15 min before TsV inoculation; propranolol (Sigma-Aldrich/Merck, USA), 300 μl in PBS, 4 mg kg$^{−1}$ i.p[49] 60 min before TsV inoculation; atropine (Sigma-Aldrich/Merck, USA) 300 μl in PBS, 2 mg kg$^{−1}$ i.p[49]. Sixty minutes before TsV inoculation; indomethacin

(Sigma-Aldrich/Merck, USA) 300 μl in Tris[hydroxymethyl]aminomethane-HCl; TRIS-HCl, pH 8.2, 2 mg kg$^{-1}$ i.p[4]. given 15 min or 60 min after TsV; prostaglandin EP2 receptor antagonist AH6809 (Cayman Chemical Company, USA), and EP4 prostanoid receptor antagonist AH23848 (Cayman Chemical Company, USA), both 5 mg kg$^{-1}$ i.p[4]. 24 h and 60 min before TsV inoculation.

**Flow cytometry.** Heart cell suspensions were obtained after tissue digestion at 37 °C for 60 min in 1 ml of heart digestion buffer (RPMI 1640, Liberase LT, Roche, Basel, Switzerland, and DNase 0.5 mg ml$^{-1}$, Sigma-Aldrich, St. Louis, MO, USA). The tissue fragments were passed through a cell strainer with 100-μm pore size (BD Biosciences, Franklin Lakes, New Jersey, USA). Next, the red blood cells were lysed, and the remaining cells were washed in PBS, centrifuged, and resuspended in RPMI 1640 containing 5% FBS. Suspensions of $1 \times 10^6$ cells from heart tissue were used for further analysis. The following antibodies were used: CD11b (clone: M1/70); CD45 (clone: 30-F11); Ly6G (clone: RB6-8C5); MHCII (clone: M5/144.15.2); F4/80 (clone: BM8); CD31 (clone: EPR17259); PDGFR-α (clone: APA5); CD14 (clone: rmC5-3); TLR4 (clone: MTS510); and TLR2 (clone: 6C2). The antibodies used for flow cytometry were purchased from eBioscience (San Diego, CA), BD Biosciences (Franklin Lakes, New Jersey, USA), or Abcam (Cambridge, UK). Data acquisition was performed using a BD LSRFortessa flow cytometer and FACSDiva software (BD Biosciences, Franklin Lakes, New Jersey, USA). In total, 1,000,000 events were acquired for samples. Data were plotted and analyzed using FlowJo software v.10.0.7 (Tree Star, Inc, Ashland, OR, USA). Gating strategies are shown in Supplementary Fig. 2.

**Quantitative PCR with reverse transcription.** Heart lysates or cardiac fibroblasts were used for RT-qPCR. RNA was extracted using RNA-extractions kits according to the manufacturer's instructions (Purelink, Ambion, USA), and the quantification was determined by fluorometric assays (Qbit, Invitrogen, CA, USA). Complementary DNA (cDNA) was synthesized from 1 μg of total RNA according to the manufacturer's protocols (High Quality cDNA Reverse Transcriptase kits, Applied Biosystems, CA, USA). A total of 50 ng of cDNA was used for RT-qPCR using Taqman primers for *Alox5, Alox5ap, Blt1, Blt2, Il1r1, Casp1, Nlrp3, Cd14, Cd36, Tlr2, Tlr4,* and *Ptger2* in a StepOne Plus machine (Applied Biosystems, USA). *Gapdh* was the reference gene used in all experiments. All reactions were performed in triplicate, and cycles were performed under the following conditions: denaturation at 95 °C for 2 min, followed by 40 cycles of 95 °C for 2 s and 60 °C for 20 s. The $2^{-\Delta\Delta Ct}$ method was performed on all samples for analysis. For the heatmap, the *n*-fold difference relative to control (all gene expressions were normalized to *Gapdh*) was used to plot the results using the *Heatmapper* platform.

**Western blotting.** Hearts from C57Bl/6 envenomed mice were removed and digested in RIPA buffer (Sigma-Aldrich/Merck, Darmstadt, Germany). Next, protein quantification of the supernatant was performed using Lowry reagents (DC™ Protein Assay, Bio-Rad, USA). The supernatants were suspended with LDS Sample Buffer (4x) (Life Technologies, USA), boiled, resolved by SDS-polyacrylamide gel electrophoresis (10–15% gel), and transferred onto a 0.22-μm-nitrocellulose membrane (GE HealthCare, USA). The membranes were blocked with Tris-buffered saline containing 0.01% Tween 20 and 5% nonfat dry milk. The monoclonal antibodies to PKA-γ clone [EP2647Y] (Abcam, USA) and PKA (phospho T197) clone [EP2606Y] (Abcam, USA) were diluted in blocking buffer for incubation. Antibody detection was done using Enhanced Chemiluminescence Reagent (GE Healthcare, USA).

**Measurement of intracellular cAMP.** For intracellular cAMP measurement, $8 \times 10^5$/well of CFs were incubated with 1 ml of DMEM serum-free with or without PGE$_2$ (10 μM) for 3 min before stimulation with TsV (50 μg ml$^{-1}$) for 5 min. Cell culture supernatants were aspired, and the cells were lysed by incubation for 10 min with 0.1 M HCl, followed by disruption using a sterile syringe. Intracellular cAMP was quantified by ELISA using an acetylation protocol according to the manufacturer's instructions (Enzo Life Sciences, NY, USA).

**Quantification of IL-1β, PGE$_2$, acetylcholine (ACh), and catecholamines.** IL-1β in the supernatant of heart homogenates or cell culture supernatants was quantified using an IL-1β ELISA kit (BD Biosciences, USA). To obtain plasma or serum, animals were anesthetized with ketamine plus xylazine, and blood was collected by cardiac puncture with heparinized or not syringes. For plasma or serum separation, the blood was centrifuged for 500×g for 10 min at 4 °C. After separation, plasma or serum was immediately frozen in liquid nitrogen until the quantification was performed. For PGE$_2$ quantification, the supernatant of heart homogenate or serum was previously treated in Sep-Pak C$_{18}$ cartridges (Waters Corp.), and the measurement was performed using the EIA assay (Enzo Life Sciences). ACh quantification was performed in the plasma using a colorimetric assay according to the manufacturer's instructions (Choline/Acetylcholine Assay kit, Abcam, USA). Briefly, free choline was measured, and, using the same sample, the acetylcholinesterase enzyme was added to the sample, and the total

choline was measured again. The total choline was subtracted from free choline, resulting in the amount of ACh in plasma samples. Plasma catecholamines (adrenaline and noradrenaline) were measured using HPLC (LC20AT-Shimadzu Prominence, Japan) coupled to an electrochemical detector (Decade Lite-Antec Scientific, Netherlands) with a 5-μm Spherisorb ODS-2 reversed-phase column (Sigma-Aldrich, USA). Acetylcholinesterase activity was determined in plasma according to the manufacturer's instructions (Acetylcholinesterase assay kit, Abcam, USA).

**Vagotomy and sham surgery.** A unilateral cervical vagotomy technique was performed. Briefly, the animals were submitted to analgesia with ketamine (0.05 mg g$^{-1}$) and xylazine (0.02 mg g$^{-1}$) (i.p.). A median cervical ventral incision was made to expose the right vagal trunk, which was then separated from the carotid artery and sectioned. In sham animals, the surgical intervention was similar to the denervated group, but without sectioning of the vagus nerve. Five days after the procedure, the animals were inoculated with a lethal dose of TsV, and after 60 min, the humanitarian endpoint, mice were euthanatized by cervical dislocation, preceded by anesthesia with ketamine and xylazine.

**Statistical analysis.** We performed Student's *t* test for two-group comparison, and one-way analysis of variance (ANOVA) followed by Bonferroni's multicomparison test for comparison of multiple groups. For survival analysis, the log-rank test was used for the determination of statistical significance. Specifically for echocardiogram analysis, paired *t* test was performed. In all tests, differences with $P < 0.05$ were considered statistically significant.

**Reporting summary.** Further information on research design is available in the Nature Research Reporting Summary linked to this article.

## Data availability

All data supporting the results reported here are available upon reasonable request to the corresponding author. Source data are provided with this paper.

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

## Acknowledgements

This work was supported by grants from Sao Paulo Research Foundation (FAPESP; grants, #2014/07125-6 and EMU #2015/00658-1 to L.H.F., #2017/02314-3 to M.B.R., and #2015/21976-1 to F.L.R. Additional support was given by the National Council for Scientific and Technological Development (CNPq) and the Coordination for the Improvement of Higher Educational Personnel (CAPES—Finance Code 001)). We also thank Faculdade de Ciências Farmacêuticas de Ribeirão Preto, Universidade de São Paulo, and Faculdade de Medicina de Ribeirão Preto, Universidade de São Paulo, for the institutional support, and the Butantan Institute for providing the scorpion venom and the antiscorpionic antiserum. Figure 6 was created using images from Servier Medical Art licensed under a Creative Commons Attribution 3.0.

## Author contributions

M.B.R., F.L.R., N.L., C.A.S., A.F.G.M., C.A.A.S., K.F.Z., C.O.S.S, S.G.R., L.B.R., and L.G.G. contributed to the data collection, analysis, and interpretation. M.B.R., F.L.R., I.C.K., S.G.R., A.K.M., L.C.C.N., H.C.S., P.C., L.G.G., and A.K. contributed to the study design and data interpretation. L.H.F. conceived and supervised the project, contributed to the study design, helped with the data interpretation, and participated in the data analysis. M.B.R., A.K., L.G.G., and L.H.F. wrote the paper. All authors read and approved the final version of the paper.

## Competing interests

The authors declare no competing interests.
