## [Peer Review File · Nature Communications]

Reviewers' Comments:

Reviewer #1:

Remarks to the Author:

Reis and colleagues determined the molecular mechanism of scorpion venom in context of inflammatory cytokine and eicosanoid kinetics with major relevance to cardiac pathology and survival. Consideration of the following points will improve the current of the manuscript. There are multiple weaknesses than strengths that needs to be considered in order to improve this current form of the manuscript.

1. Authors needs to re-consider cytokine versus eicosanoids potency, supplementary figure 1 clearly indicate that PGE2 levels reached to peak in thoughts after 60 min but IL-1b in hundreds thus needs caution to interpret these results particularly driven by eicosanoids rather than cytokines (consider comparison of Y axis of PGE2 and IL-1b).
2. As suggested in the title provide the kinetics of IL-1b, PGE2 and acetylcholine on the panel to understand the molecular dynamics of three mediators in scorpion toxicity.
3. Line – 19 – Author hypothesized neuroimmune interaction with cardiac toxicity, however the cardiac toxicity explained with limited focus on neuroimmune.
4. Figure 1. K, L and M clearly indicate that PGE2 is primary indicator of toxicity in local and systemic manner, discuss the source and mechanism of initiation in context of cellular source.
5. Page 7 figure 8. What are PGE2 levels in 1L1Br null mice? In heart and circulation
6. Compress heart function data in meaningful way in order to have better readership particularly to students or non-experts.
7. After addition of PGE2 – many of the cell culture experiments are performed after 24 hours (extended data fig 4) in that case the half-life of PGE2 is extremely less/critical and the effect is relative from other corollary activation, needs precision in the interpretation of results.
8. Cardiac ultrasound images are not providing the input since the heart rate is lowered within 60 minutes due to toxin.
9. Echocardiography precision is unclear because if the mice dormant stage the function will be unclear particularly in short period of time due to lower heart rate. On the ultrasound machine it requires 10-15 minutes to get physiology relevant heart functional data.
10. Figure PGE2-EP4 axis description is unclear because EP4 mRNA expression and activation is not presented, thus needs caution for this interpretation.
11. IL1b receptor null mice survive better as per figure 1, and with DEX mortality is prevented in this case what are PGE2 levels and is there activation of EP4 either in C57BL/6 mice or IL1b null mice.
12. Extended data figure 3. Add precise marking in the figure for review information to indicate mitochondrial tumefaction.
13. Heart failure is secondary mechanism, the primary mechanism of initiation of inflammation signaling is unclear in the presented paper.

Reviewer #2:

Remarks to the Author:

The manuscript by Reis et al examines the effect of scorpion venom on cardiac dysfunction and mortality in mice; providing a link between interleukin-1 mediated PGE2 production and the dysfunction induced by acetylcholine. The authors show that both heart failure and mortality were abolished by treatment with dexamethasone or in IL1r-/- mice leading them to suggest that administration of dexamethasone might reduce the risk of death in patients. In general, the manuscript is interesting and the methodology is adequately written so that the study could be reproduced. However, I have the following concerns and critique:

1. The formatting of the manuscript is confusing and difficult to follow. Results of the study are interspersed with what appears to be the introduction (page 2) and extended data figures are presented which appear to be different that the regular figures or the supplemental figures. This

reviewer suggests that it would be easier to read if all supplemental material were in one place.

2. At the conclusion of the manuscript, the authors suggest the administration of dexamethasone to patients after scorpion bites. However, this is not a new concept. Another group (Malaque et al, Intensive Care Med Exp. 2015 Dec; 3: 28) from Sao Paulo School of Medicine, Brazil reported the "role of dexamethasone in scorpion venom-induced deregulation of sodium and water transport in rat lungs" and reported that in Brazil, it is common to administer corticosteroids prior to the administration of anti-venom. The mechanistic aspects of the manuscript are novel and this reviewer suggests that more emphasis is made on this point.

3. In the extended data figure 1 legend, it would be useful to know at what time after venom the cardiac gene expression profiling was performed.

4. Some of the data raises questions that need addressing. For example, in extended data figure 2, panel b shows a substantial increase in PGE2 levels in cardiac fibroblasts after stimulation with TsV. However, panel g shows no increase in cAMP after TsV stimulation of cardiac fibroblasts. These discrepancies need to be discussed. If PGE2 is acting via either its EP2 or EP4 receptor, one would expect an increase in cAMP. Was a phosphodiesterase inhibitor added to the media to prevent its breakdown? Additionally, panel I shows that TsV administration to whole animals increases PKA in homogenized hearts.

5. The link between release of PGE2 mainly by the cardiac fibroblast and its effect on cardiac myocyte physiology is not well elucidated and is not clear from the figures. This aspect should be discussed further. Additionally, it is stated that only cardiac fibroblasts release IL-1b. The amount released from the cardiac myocytes was undetectable using an ELISA kit but is probably not zero so the limits of detection for the assay should be given.

6. Administration of atropine did not affect levels of IL-1b in response to TsV but markedly improved the clinical score and survival. The graphs for cardiac output, stroke volume and ejection fraction (Figure 2 i) are misleading in that they only show the data for basal and TsV + atropine. TsV alone should be shown on these graphs so that one can ascertain whether TsV + atropine is different than TsV alone.

7. The electron microscopy and discussion of mitochondrial dysfunction/ultrastructural abnormalities does not add to the manuscript as there is no quantification of the data and no evidence presented of alterations in mitochondrial energetics. This part could be removed from the manuscript.

8. The legends for the survival curves presented in figures 1 e and 1h state that "n=10 or n=5, representative of two experiments". What does this mean? Cannot the entire data be presented?

9. In mice treated with atropine before inoculation with TsV, there was a substantial reduction in clinical score and survival was markedly improved, yet, atropine did not affect IL-1b concentrations in either the heart or the lungs. How does this fit with the author's hypothesis that TsV-induced cardiac dysfunction and mortality depend on IL-1R mediated PGE2 production and acetylcholine release.

10. There is really very little discussion in the manuscript except to say that dexamethasone would prove useful to block the inflammatory mediators and release of acetylcholine in situations where there may be a delay in the administration of antivenom and increased risk of death. The discussion should be enhanced and widened in scope.

11. As a minor point, the references need to contain titles for the articles.

Reviewer #3:

Remarks to the Author:

Brazil had 99 deaths and more than 90,000 scorpion accidents in 2018, according to the Ministry of Health. More than 40,000 of them occurred in the Southeast region of Brazil.

This manuscript represents an extensive and detailed approach for establishment of a murine model of severe scorpion envenomation for the elucidation of the molecular mechanisms that lead cardiac dysfunction and mortality.

Data are consistent, and authors propose an additional therapeutic intervention with high dose of DEX to block the production of inflammatory mediators to prevent heart failure.

Specific comments:

A definitive proof of concept is to compare the effectiveness of DEX with antivenom serum to prevent heart failure, or even to evaluate the additive effect of DEX to antivenom serum.

Reviewer #4:

Remarks to the Author:

The manuscript by Reis et al. describe a mechanism of scorpion venom (TsV) induced death due to cardiac dysfunction. The authors propose a pathway involving IL-1(beta), prostaglandins, and Acetylcholine.

Importantly the authors identify IL1beta as a critical mediator in this process, and further identify the ability of dexamethasone to reduce mortality as a treatment modality. Although interesting there are some issues that would in my view prevent publication at this time. In particular, the proposed mechanism of cAMP enhanced Ach release from neuronal terminals seems underdeveloped or lacking evidence. As currently written the manuscript is not reflective on prior studies nor is there any real discussion or framing of the current results into any broader context. These issues would prevent publication in my opinion.

Major points:

The organization of the manuscript is odd. I'm at a loss to understand why there are 4 extended data figure. The interweaving of figures and figure legends with main text makes this hard to follow.

Are the components of the TsV causing different aspects of the observed phenomenon?

Extended data 1:

How does 1d fit in with prior reports using TsV showing increased blood neutrophils only after 360 min? PMID: 21893076

In extended data 1 "h" and "I", is the reduction in frequency of macrophages simply due to the increased frequency of neutrophils? Is the absolute number different? I'm also not sure I see what the importance of this flow data is.

Please remove the heatmap. Unless I am mistaken this is from qPCR data? This should be presented in another format, violin plot, bar graph so that one can see the variation in the data.

I could not tell if in extend fig 1L the heart data was perfused heart or if there was still blood in the tissue at the time of analysis. These data would also indicate that the heart is a major target of TsV and a major producer of PGE2? Have other organs be checked, presumably the lung is also a source? Is one site more important than the other? Please clarify.

Extended data 2:

Please change "c" to a graph type that allows for readers to discern the variation in your experiments. The description of this in the figure legend is 3/wells per condition. Is this 3 wells per condition?

Is the osmolarity of the NaCl solution the same as the KCl in "e"?

AH6809 is a EP1/EP2 antagonist, but does not effect EP3 or EP4 this does not come across in the

text where it is described as a "PGE2 antagonist (compound AH 6809),...". This should probably be revised to "...a PGE2 EP1/2 receptor antagonist..." or similar to highlight that it is not a pan antagonist.

In "f" Why is there no TsV + Ah6809 group? I also didn't see a PGE2 control group, so is the PGE2+ Tsv really increased or is this simply the effect of PGE2 alone? Similarly why is there no AH6809 + TsV group?

In "g" if TsV increases PGe2 production to drive increased cAMP, why isn't cAMP increased by TsV alone? Is this at the limit of detection of the assay?

"j" I didn't see direct evidence presented that indicates that PKA activates Nf-kb signaling in this system. This would be important as several other cAMP/PKA inhibits NF-kb in some cell types. I also do not understand why PGE2 seems to be coming from the lung in this diagram? Does this indicate that the predominant source is not the heart tissue itself?

Figure 2

Is it odd that Propanolol had been previously reported to prevent selected aspects (contractile force coronary flow) of TsV y induced cardiac dysfunction? (see "EFFECTS OF TZTYUS SERRULATUS SCORPION VENOM AND ONE OF ITS PURIFIED TOXINS (TOXIN y) ON THE ISOLATED GUINEA-PIG HEART" Silveira et al. 1991), and yet here it seems to not have had any effect on the parameters measured?

In "i" it seems as though the TSv vehicle groups are not included. Is there a reason for this? Does "L" suggest that Dex isn't a great treatment, it seems that it would need to be given within 15 mins of being stung. Perhaps the therapeutic statements should be re-written to downplay this given these data.

In "m", is the point that the lung is the most important source of PGe2?

Based on the previous data presented, I'm surprised that PGE2 administration didn't increase Ach in the Il-1R-/- mice. Does this not indicate that the model presented where PGE2 acts via EP1/2 and (now ep4 as well based on the AH23848 data) increased ACh release is potentially not complete or incorrect? Is this a dose or time issue? Since mPGE2s inhibition in "O" seems to reduce Ach release induced by TsV?

The model although nicely drawn does not seem to consider that other aspects of neurotransmission could be affected by the toxin. I don't see any evidence to suggest that neural tone isn't changed for example. In other words, how can one be certain that the increase in ACh isn't due to an alteration of the input to the heart or circuitry at the level of stellate ganglion or any circuit in a variety of tissues? To state that since cAMP can increase ACh release in neurons and you have increased PGE2 which can increase cAMP is a real stretch. I didn't see any measures of cAMP in neurons that could substantiate this.

Does TsV cause cell death? I didn't see the use of live/dead indicator in the flow experiments. Seems as though there are several manuscripts describing this at least in vitro.

The discussion is not very substantial. This would in my view need to be improved to be sure the manuscript highlights how this fits into prior literature or related literature.

Minor points:

Use continuous line numbers.

In ext data2, I think the axis on "d" should read "% CD14+ cells of CD31- PDGFR+" and similar for TLR2 and TLR4.

On page 4 ln 22 should read "with an elevated concentration..."

Page 6 line 20 seems an odd choice of words "untreated-WT envenomed" should maybe be vehicle treated?

Page 11 line 26 "The data express mean..." should be "The data are expressed as..."

Original article: “Interleukin-1 receptor-induced PGE₂ production controls acetylcholine-mediated cardiac dysfunction and mortality during scorpion envenomation”

Reviewers' comments:

Reviewer #1 (Remarks to the Author):

Reis and colleagues determined the molecular mechanism of scorpion venom in context of inflammatory cytokine and eicosanoid kinetics with major relevance to cardiac pathology and survival. Consideration of the following points will improve the current of the manuscript. There are multiple weaknesses than strengths that need to be considered in order to improve this current form of the manuscript.

1. Authors needs to re-consider cytokine versus eicosanoids potency, supplementary figure 1 clearly indicate that PGE₂ levels reached to peak in thoughts after 60 min but IL-1 β in hundreds thus needs caution to interpret these results particularly driven by eicosanoids rather than cytokines (consider comparison of Y axis of PGE₂ and IL-1 β).

Reply: Thank you for your observation. In a previous publication by Zoccal et al. (2016), we described the kinetics of IL-1 β and PGE₂ production in the lungs after scorpion envenomation. As demonstrated in this paper, both mediators are inducible by TsV, and our results suggest that PGE₂ is produced in two waves. The first wave originates from resident cells and triggers NALP3 inflammasome activation and IL-1 β release. The IL-1 β released consequently amplifies the production of PGE₂. Thus, there is a looping effect and both can amplify each other's production.

2. As suggested in the title provides the kinetics of IL-1 β , PGE₂ and acetylcholine on the panel to understand the molecular dynamics of three mediators in scorpion toxicity.

Reply: The kinetics of all the mediators mentioned are already shown in the manuscript, in Extended Data 1 (PGE₂ and IL-1 β) and Fig 2.e (acetylcholine).

3. Line – 19 – Author hypothesized neuroimmune interaction with cardiac toxicity, however the cardiac toxicity explained with limited focus on neuroimmune.

Reply: Thank you for your observation. If we correctly understood, the reviewer is inquiring whether cardiac toxicity is regulated by neuroimmune interactions. We performed new experiments that address this in the new version of this manuscript (Line 171).

In the old version of the manuscript, we demonstrated that atropine, as a muscarinic antagonist, reverted venom-induced cardiovascular dysfunction. As demonstrated in the new version of the manuscript, we also performed a cervical vagotomy and notably observed that this surgical vagal blockage preserved cardiovascular functions as well as the clinical score after venom inoculation (Fig 5). Overall, our results reveal a new parasympathetic neuro-immune pathway involved in the “neurotransmitter storm” that is described in venom-related clinical articles (Cupo et al., 2015; Isbister et al., 2014).

4. Figure 1. K, L and M clearly indicate that PGE₂ is primary indicator of toxicity in local and systemic manner, discuss the source and mechanism of initiation in context of cellular source.

Reply: We believed that this question is related to Extended Figure 1. As mentioned and has been previously described by our research group in a *Nature Communications* publication (Zoccal et al., 2016), after scorpion envenomation, the main source of PGE₂ are resident and recruited macrophages in the lungs. Scorpion venom (TsV) also up-regulates COX-2 in macrophages and increases PGE₂ production, thereby suggesting that the macrophages release PGE₂. However, we cannot discard the contribution of others cells, such as peripheral mononuclear cells, as recently described (Zoccal et al., 2018). Additionally, articles in the literature report that macrophages are principal cells that produce this lipid mediator (Kalinski P, 2012). Although this question is relevant, our aim in this manuscript is not to identify the particular cell type involved, but to demonstrate that the lungs are the main source of lipid mediators that disturb heart functions via blood access. We are appreciative of this observation and included it in the “Discussion” section. We assume that the possible source(s) of PGE₂ are important to identify, and thus we added the following sentence to the text: “We have previously reported that after scorpion envenomation, the main source of PGE₂ are resident macrophages in the lungs, but the contribution

of other peripheral mononuclear cells cannot be discarded (Zoccal et al, 2016; 2018). However, further studies are necessary to better assess this important aspect.” (Line 256).

5. Page 7 figure 8. What are PGE₂ levels in IL1Br null mice? In heart and circulation

Reply: We wish to thank the reviewer for this important observation. Data are now included in the new version of our manuscript (Line 147) and described in Figure 3 m-n and its respective legend.

6. Compress heart function data in meaningful way in order to have better readership particularly to students or non-experts.

Reply: Thank you for your suggestion, as it is important that this manuscript is clear for readers who are not familiar with specific techniques, such as the echocardiogram. To clarify the cardiac function data, the sentence “The echocardiographic analysis showed that only untreated-WT envenomed mice presented disturbances of the systolic and diastolic movements of the left ventricle walls, as reflected by reductions in the ejection fraction (EF), stroke volume (SV), and cardiac output (CO)” was replaced by “The echocardiographic analysis showed that only untreated-WT envenomed mice presented cardiac dysfunction, as estimated by reductions in the stroke volume (SV; the volume of blood pumped out of the left ventricle of the heart during each systolic cardiac contraction), ejection fraction (EF; measurement, expressed as a percentage, of how much blood the left ventricle pumps out with each contraction) and cardiac output (CO; the blood volume the heart pumps through the systemic circulation over a period measured in liters per minute.).” (Line 137)

7. After addition of PGE₂ – many of the cell culture experiments are performed after 24 hours (extended data fig 4) in that case the half-life of PGE₂ is extremely less/critical and the effect is relative from other corollary activation, needs precision in the interpretation of results.

Reply: Thank you for this important question. However, we do not agree with this observation. PGE₂ (as others lipid mediators), once binded to its receptor (EP2-EP4), induces intracellular reprogramming events, as demonstrated by others and in our laboratory. Based on our experience, we know that continuous addition of PGE₂ is not necessary to induce cell stimulation. Studies from others and our own laboratory

describe very robust effects of PGE₂ using the same methodology (Zoccal et al., 2016, Figure 5; Zoccal et al., 2018, Figure 3; Pelletier et al., 2001, Figure 8; Scandella et al., 2002, Figure 1; Dahiya et al., 2010, Figure 3; Sheppe et al., 2018, Figure 4; Hinz et al., 2010, Figure 2).

Extended Figure 4 (now Figure 2) shows the effects of PGE₂ in amplifying inflammasome activation by increasing cAMP-PKA. This result was expected once it was demonstrated that, in macrophages, PGE₂ can increase IL-1 β release in vitro using the same protocol (24 h).

8. Cardiac ultrasound images are not providing the input since the heart rate is lowered within 60 minutes due to toxin.

Reply: We apologize, but the representative images are from echocardiography in the M mode, and not from ultrasound images. The images were indeed captured at 60 min after envenomation, when the heart rate and cardiac function were seriously damaged. As such, we thank you for your comment. The representative images from the echocardiography were obtained throughout the cardiac cycle from the short-axis or the motion-imaging mode (M-mode). In this figure, our goal was to qualitatively demonstrate the effects of scorpion venom (TsV) on the left ventricle internal diameter and the left ventricle posterior and anterior wall diameters. Therefore, we did not expect the input observed in this Figure.

9. Echocardiography precision is unclear because if the mice dormant stage the function will be unclear particularly in short period of time due to lower heart rate. On the ultrasound machine it requires 10-15 minutes to get physiology relevant heart functional data.

Reply: We agree with the reviewer and thank them for highlighting this point. In fact, to obtain consistent, reproducible echocardiographic data, aside from obtaining good images, the condition of the animals should also be controlled. The mouse body temperature should be carefully monitored, the echo measurement time should be similar after anaesthesia, and the heart rate should be controlled, since cardiac function is closely related to heart rate. Due to the limitations of echocardiography in conscious mice, anaesthesia is frequently used in murine echocardiography. Once the animal has been properly prepared and good images are obtained, both systolic and diastolic cardiac function can be accurately measured and compared to monitor

cardiac pathophysiology, as well as the effectiveness of any intervention. In the present study, the above parameters were strictly controlled and the decreased heart rate was a direct effect of our intervention and was compensated by larger recording windows. We apologize for our mistake in the explanation of this experiment. We did not specify it in an extensive manner, due the prior format of the manuscript. We included more information in the text: “Envenomed mice had their echocardiograph examination performed for 15 min, starting 45 min after envenomation and ending 60 min after TsV.” (Line 432)

10. Figure PGE₂-EP4 axis descriptions is unclear because EP4 mRNA expression and activation is not presented, thus needs caution for this interpretation.

Reply: We thank the referee for this suggestion. We believed that, in this case, it is not necessary to show increased mRNA, as this technique does not reflect the real expression of the protein. On the other hand, using a specific EP4 antagonist (Coleman et al., 1994), it was possible to reverse the phenotype observed. Herein, pre-treatment with the EP4 antagonist prevented the increase of ACh induced by a lethal dose of venom (Figure 5I). In the literature, the use of selective antagonist treatment is the most specific tool to demonstrate the mechanisms by which PGE₂ induces biological activities (Zoccal et al, 2016, Sanches et al., 2002, Ma et al., 2006).

11. IL1b receptor null mice survive better as per figure 1, and with DEX mortality is prevented in this case what are PGE₂ levels and is there activation of EP4 either in C57BL/6 mice or IL1b null mice.

Reply: Thank you for the very important question. In the new version of the manuscript, the PGE₂ concentration in DEX-treated mice was included in Figure 3h-i, Line 429.

12. Extended data figure 3. Add precise marking in the figure for review information to indicate mitochondrial tumefaction.

Reply: Thank you for the suggestion. We removed this figure as suggested by other reviewers due to lack of more evidences about this event.

13. Heart failure is secondary mechanism; the primary mechanism of initiation of inflammation signalling is unclear in the presented paper.

Reply: Thank you for the excellent question. We believe that this conclusion indeed sounds confusing because the length of the first version of the manuscript as a “Letter” limits the explanation. In the new version of the manuscript, we better discussed and elucidated this aspect. We suggest that the inflammation occurs systemically, and immune cells, such as resident macrophages or recruited cells to the lungs, pancreas, heart, and blood, are the main source of mediators. Macrophages and neutrophils are sources of lipid mediators, as demonstrated before (Dennis & Norris, 2015; Khanapure et al., 2007). After the initiation of the inflammatory process, PGE₂ was found to act systemically and increase acetylcholine release (Figure 5j-m and Line 171).

Reviewer #2 (Remarks to the Author):

The manuscript by Reis et al examines the effect of scorpion venom on cardiac dysfunction and mortality in mice; providing a link between interleukin-1 mediated PGE₂ production and the dysfunction induced by acetylcholine. The authors show that both heart failure and mortality were abolished by treatment with dexamethasone or in IL1r^{-/-} mice leading them to suggest that administration of dexamethasone might reduce the risk of death in patients. In general, the manuscript is interesting and the methodology is adequately written so that the study could be reproduced. However, I have the following concerns and critique:

1. The formatting of the manuscript is confusing and difficult to follow. Results of the study are interspersed with what appears to be the introduction (page 2) and extended data figures are presented which appear to be different that the regular figures or the supplemental figures. This reviewer suggests that it would be easier to read if all supplemental material were in one place.

Reply: We apologize for that. We were informed that the transfer process from *Nature Medicine (Letter)* to *Nature Communications (Article)* did not require adjustments to the manuscript. We believe that this information should have been passed to reviewers. The way the manuscript is written is because the letter format for publication in *Nature Medicine* is very concise. We are sending a more appropriately structured manuscript with an extensive discussion and improved organization with the updated manuscript.

2. *At the conclusion of the manuscript, the authors suggest the administration of dexamethasone to patients after scorpion bites. However, this is not a new concept. Another group (Malaque et al, Intensive Care Med Exp. 2015 Dec; 3: 28) from Sao Paulo School of Medicine, Brazil reported the “role of dexamethasone in scorpion venom-induced deregulation of sodium and water transport in rat lungs” and reported that in Brazil, it is common to administer corticosteroids prior to the administration of anti-venom. The mechanistic aspects of the manuscript are novel and this reviewer suggests that more emphasis is made on this point.*

Reply: This conclusion must be done carefully. The protocols of DEX administration must be performed in case of administration of the highest doses of serum to prevent serum sickness. This is not done routinely. Nevertheless, the manuscript mentioned above does not reference the effects of DEX in terms of heart function. We included a better statement and protocols regarding this question in the “Discussion” section of the formatted manuscript.

3. *In the extended data figure 1 legend, it would be useful to know at what time after venom the cardiac gene expression profiling was performed.*

Reply: We apologize for that. The information was included as suggested.

4. *Some of the data raises questions that need addressing. For example, in extended data figure 2, panel b shows a substantial increase in PGE₂ levels in cardiac fibroblasts after stimulation with TsV. However, panel g shows no increase in cAMP after TsV stimulation of cardiac fibroblasts. These discrepancies need to be discussed. If PGE₂ is acting via either its EP2 or EP4 receptor, one would expect an increase in cAMP. Was a phosphodiesterase inhibitor added to the media to prevent its breakdown? Additionally, panel I shows that TsV administration to whole animals increases PKA in homogenized hearts.*

Reply: In fact, our results raise questions on the role of PGE₂ in cAMP induction. In the new version of the manuscript, we included a possible explanation in the “Discussion” section (Line 256) with a new reference (Delaunay et al., 2020).

Related to in vivo treatment, we consider that the positive results relate to the amount of PGE₂ administered to the mice. The animals were treated with 1 mg/kg (or 35 μM/mice, which is the same high dose that is described in the literature (Long et al., 1991). To summarize, our results demonstrate that a high dose of PGE₂ is necessary

to induce cAMP. As we discussed in lines 256, this may represent a mechanism to protect the heart from fibrosis.

We did not treat the cells with phosphodiesterase inhibitor.

5. The link between release of PGE₂ mainly by the cardiac fibroblast and its effect on cardiac myocyte physiology is not well elucidated and is not clear from the figures. This aspect should be discussed further. Additionally, it is stated that only cardiac fibroblasts release IL-1b. The amount released from the cardiac myocytes was undetectable using an ELISA kit but is probably not zero so the limits of detection for the assay should be given.

Reply: The question is very interesting. However, in this manuscript, our main objective is to elucidate the mechanism by which TsV induces cardiac dysfunction. Thus, independent of the source of PGE₂ and IL-1 β (local or systemically), the main role of the lipid mediator is to amplify the cytokine production, and both contribute to ACh release and induce cardiac alterations. As discussed, local and systemic mediators play an important role in scorpion envenomation. In the next project, we will investigate in more detail the relationship between PGE₂, IL-1 β , and mitochondrial tumefaction.

6. Administration of atropine did not affect levels of IL-1b in response to TsV but markedly improved the clinical score and survival. The graphs for cardiac output, stroke volume and ejection fraction (Figure 2 i) are misleading in that they only show the data for basal and TsV + atropine. TsV alone should be shown on these graphs so that one can ascertain whether TsV + atropine is different than TsV alone.

Reply: Thank you for the question. In the new version of the manuscript, these results are shown in Fig.5.

7. The electron microscopy and discussion of mitochondrial dysfunction/ultrastructural abnormalities does not add to the manuscript as there is no quantification of the data and no evidence presented of alterations in mitochondrial energetics. This part could be removed from the manuscript.

Reply: We appreciate your observation and agree that the investigation of the energetic function of mitochondria in scorpion envenomation is missing. Although we suppose that alteration in the energetic the balance of myocytes occurs in scorpion

envenomation, we will investigate this matter in a future project. As suggested, we removed the mitochondrial results.

8. *The legends for the survival curves presented in figures 1 e and 1h state that “n=10 or n=5, representative of two experiments”. What does this mean? Cannot the entire data be presented?*

Reply: We apologize for the mistake. In the new version of the manuscript, we corrected the number of animals in terms of the survival curves and clinical scores.

9. *In mice treated with atropine before inoculation with TsV, there was a substantial reduction in clinical score and survival was markedly improved, yet, atropine did not affect IL-1b concentrations in either the heart or the lungs. How does this fit with the author’s hypothesis that TsV-induced cardiac dysfunction and mortality depend on IL-1R mediated PGE₂ production and acetylcholine release.*

Reply: Our main hypothesis is that the immune response regulates the autonomic response, rather than the opposite. Our results support our hypothesis, as explained here. Supplementary Fig 1 shows that the IL-1R-PGE₂ axis occurs very early after envenomation, but ACh release begins later, at 60 min after the venom is injected. As discussed in the new version of the manuscript, our results suggest that, after the venom triggers the release of inflammatory mediators (that induce ACh production), whether the venom is degraded and/or neutralized by the antiserum is irrelevant. Moreover, our results clearly demonstrate that the anti-inflammatories DEXA or Indomethacin inhibit the release of inflammatory mediators and consequently ACh release. Thus, as expected, the results demonstrate that atropine treatment does not decrease IL-1 β , since atropine antagonizes muscarinic receptors. We assume that this may be why some patients still have cardiac alterations even after serum administration.

10. *There is really very little discussion in the manuscript except to say that dexamethasone would prove useful to block the inflammatory mediators and release of acetylcholine in situations where there may be a delay in the administration of antivenom and increased risk of death. The discussion should be enhanced and widened in scope.*

Reply: As explained previously, this limitation occurred due to transferring the manuscript from Nature Medicine, for which we submitted the manuscript in a letter format. We updated a new version of the manuscript, which includes an extensive “Discussion” section.

11. As a minor point, the references need to contain titles for the articles.

Reply: We reformulated this section.

Reviewer #3 (Remarks to the Author):

Brazil had 99 deaths and more than 90,000 scorpion accidents in 2018, according to the Ministry of Health. More than 40,000 of them occurred in the Southeast region of Brazil. This manuscript represents an extensive and detailed approach for establishment of a murine model of severe scorpion envenomation for the elucidation of the molecular mechanisms that lead cardiac dysfunction and mortality. Data are consistent, and authors propose an additional therapeutic intervention with high dose of DEX to block the production of inflammatory mediators to prevent heart failure.

Specific comments:

A definitive proof of concept is to compare the effectiveness of DEX with antivenom serum to prevent heart failure, or even to evaluate the additive effect of DEX to antivenom serum.

Reply: Thank you for the suggestion. We performed the suggested experiments and it improved our manuscript. The results are shown in Figure 4.

Reviewer #4 (Remarks to the Author):

The manuscript by Reis et al. describes a mechanism of scorpion venom (TsV) induced death due to cardiac dysfunction. The authors propose a pathway involving IL-1(beta), prostaglandins, and Acetylcholine. Importantly the authors identify IL1beta as a critical mediator in this process, and further identify the ability of dexamethasone to reduce mortality as a treatment modality. Although interesting there are some issues that would in my view prevent publication at this time. In particular, the proposed mechanism of cAMP enhanced Ach release from neuronal

terminals seems underdeveloped or lacking evidence. As currently written the manuscript is not reflective on prior studies nor is there any real discussion or framing of the current results into any broader context. These issues would prevent publication in my opinion.

Replying the cAMP question: We are sorry for our distraction in this issue. We believed that this point sounds confusing because the length of our first version of the manuscript sent as a “Letter”, with limited description of our results and discussion. In the new version of the manuscript, we improved all these aspects. Below we present the answers.

In the **Figure 2h** of new version, we in fact demonstrated an increase of cAMP produced by **cardiac fibroblasts** stimulated *in vitro* with TsV+PGE₂. Note that cAMP is not increased in presence of TsV alone. These results are similar to that published before by our group. In 2016, we revealed in macrophages that IL-1 β production is induced via EP2/4 prostanoid receptors, via cAMP-PKA-NF κ B-dependent pathway (Zoccal et al, *Nat Commun*, 2016).

In the **Figure 5I**, “*in vivo*” experiment, we are showing that indomethacin inhibits the systemic acetylcholine release induced by TsV (TsV-envenomed mice); and to confirm the importance of PGE₂ in the release of acetylcholine *in vivo*, we inoculated TsV in IL-1r^{-/-} mice in presence and absence of PGE₂ and EP antagonists. We observed that only in presence of PGE₂ occurs systemic acetylcholine release after venom administration. Reminding, we demonstrated before that IL-1r^{-/-} mice are deficient in eicosanoids production (Zoccal et al., 2016, *Nat Commun*).

The participation of cAMP in acetylcholine release is an interesting to point, and classical studies in the literature have demonstrated that prostaglandins increase cAMP, that in its turns induces acetylcholine release, including in peripheral neurons. We did not discuss it in the last version of our manuscript, since it was not our focus. Below are some references demonstrating the involvement of cAMP in acetylcholine release.

Takeuchi T, Hata F, Yagasaki O. Role of cyclic AMP in prostaglandin-induced modulation of acetylcholine release from the myenteric plexus of guinea pig ileum. *Jpn J Pharmacol*. 1992 Dec; 60(4): 327-33.

Cheng JT, Shinozuka K. Prostaglandin E2 induced the cyclic AMP-dependent release of acetylcholine in circular muscles of the isolated guinea pig ileum. *Neurosci Lett*. 1987 Dec 29; 83(3): 293-7.

Kalix P. Prostaglandin E₁ raises the cAMP content of peripheral nerve tissue. *Neurosci Lett*. 1979 May; 12(2-3): 361-4.

Yau WM, Dorsett JA, Youther ML. Stimulation of acetylcholine release from myenteric neurons of guinea pig small intestine by forskolin and cyclic AMP. *J Pharmacol Exp Ther.* 1987 Nov; 243(2): 507-10.

Reese JH, Cooper JR. Stimulation of acetylcholine release from guinea-pig ileal synaptosomes by cyclic nucleotides and forskolin. *Biochem Pharmacol.* 1984 Oct 1;33(19): 3007-11.

Major points:

The organization of the manuscript is odd. I'm at a loss to understand why there are 4 extended data figure. The interweaving of figures and figure legends with main text makes this hard to follow. Are the components of the TsV causing different aspects of the observed phenomenon?

Reply: We apologize for that. When the manuscript was transferred from *Nature Medicine* (submitted as a letter), nothing was informed about the necessity to rewrite the manuscript. Thus, unfortunately, you read a very concise manuscript, and should have been informed of this prior to evaluation. The new version of the manuscript is appropriate for *Nature Communications*.

Extended data 1:

How does 1d fit in with prior reports using TsV showing increased blood neutrophils only after 360 min? PMID: 21893076

Reply: Thank you for the question. The aforementioned article used the Swiss strain and s.c. route to stimulate mice with 200 µg/kg of TsV, and C57Bl/6 mice that were inoculated i.p. with 180 µg/kg of venom were used. Differences between strains are well documented by previous studies (Padilla et al., 2003), and we believed that the differences in the mice strains might explain the differences. Moreover, the intraperitoneal route, compared to the s.c. route, favors the bioavailability of the venom. Additionally, we previously demonstrated that i.p. inoculation of TsV (or its toxins) increases blood cells and induces lung and local inflammation. Please see Zoccal et al., 2016, *Nature Communications*, supplementary 3); and Zoccal et al., 2019, *Front Pharmacol*; Zoccal et al., 2013, *Toxicon*.

In extended data 1 “h” and “I”, is the reduction in frequency of macrophages simply due to the increased frequency of neutrophils? Is the absolute number different? I’m also not sure I see what the importance of this flow data is.

Reply: The data shown in the new Figure 1h-l reflects the relative recruitment of neutrophils to cardiac tissue, with consequently a relative decrease in macrophages. Note that the total number of leukocytes in the heart did not increase 1 h after TsV. This premature increase in neutrophils to the heart was expected and demonstrates that TsV induces acute heart inflammation, similar to that observed in the lung and peritoneal cavity. Please see our previous publications: Pessini et al., 2003; Zoccal et al., 2016

Please remove the heatmap. Unless I am mistaken this is from qPCR data? This should be presented in another format, violin plot, bar graph so that one can see the variation in the data.

Reply: We apologize, but we consider the heat map to be an excellent form to observe the alterations after envenomation. To clarify our results, we also included two tables with the data. Please see supplementary Tables 1 and 2.

I could not tell if in extend fig 1L the heart data was perfused heart or if there was still blood in the tissue at the time of analysis. These data would also indicate that the heart is a major target of TsV and a major producer of PGE₂? Have other organs be checked, presumably the lung is also a source? Is one site more important than the other? Please clarify.

Reply: The heart was perfused, as described in the “Methods” section. During envenomation, the greater concentration of TsV is detected in the kidney and lungs, followed by the liver and then the heart (Revelo et al., 1996). In addition to the fact that the lungs received a quicker and higher concentration of TsV, the lungs have a large number of alveolar macrophages, known to be major producers of lipid mediators (Thomas and Peters-Golden, 2007). Related to the origin of PGE₂, we previously demonstrated that the lung produces a concentration of 2-4 ng/mL (Zoccal et al., 2016, Nature Communications). Moreover, in this manuscript, we demonstrate, as observed in Figure 1o and Figure 3m, the concentration of PGE₂ that is produced by the heart to be 3-4 pg/mL. In the new version of the manuscript, we emphasize the role of the lung as the main source of PGE₂.

Extended data 2:

Please change “c” to a graph type that allows for readers to discern the variation in your experiments. The description of this in the figure legend is 3/wells per condition. Is this 3 wells per condition?

Reply: We did not remove the heat maps once we believe that this way to show gene expression is very conclusive. Instead, we added Supplementary table with values of gene expression for further analysis.

Is the osmolarity of the NaCl solution the same as the KCl in “e”?

Reply: Yes, it is.

AH6809 is a EP1/EP2 antagonist, but does not effect EP3 or EP4 this does not come across in the text where it is described as a “PGE₂ antagonist (compound AH 6809),...”. This should probably be revised to “...a PGE₂ EP1/2 receptor antagonist...” or similar to highlight that it is not a pan antagonist.

Reply: Thank you for your observation. We highlighted this information in the new version of the manuscript.

In “f” Why is there no TsV + Ah6809 group? I also didn’t see a PGE2 control group, so is the PGE2+ Tsv really increased or is this simply the effect of PGE2 alone? Similarly why is there no AH6809 + TsV group?

Reply: In Zoccal et al. 2016, we investigated the relationship between PGE2 and IL-1 β , and demonstrated that PGE2 is the first signal for inflammasome activation. As previously suggested, the release of PGE2 is induced by TsV and occurs in two waves. Resident cells that are stimulated with TsV release the first wave of PGE2 at low concentrations. The low amount of PGE2 binds to EP2/EP4 receptors and induces cAMP, which activates NF-kB and NLRP3 activation via PKA, thus releasing IL-1 β . Moreover, IL-1 β released via IL-1R amplifies the production of PGE₂ and LTB₄. Since we previously knew this mechanism (Zoccal et al., 2016), the present investigation (shown in Figure 2f of the first version of the manuscript) only aimed to

confirm that, in cardiac fibroblast, the lipid PGE₂ via EP2 receptor also potentiates the release of IL-1 β . Moreover, in the previous version of Fig. 2a, TsV is shown to induce PGE₂ release by cardiac fibroblasts. In addition, in the new version of Fig. 5, our results show that PGE₂, via EP2/EP4 receptors, is necessary for TsV-induced ACh release.

In “g” if TsV increases PGE₂ production to drive increased cAMP, why isn’t cAMP increased by TsV alone? Is this at the limit of detection of the assay?

Reply: Please see the response to question 4 for Reviewer 2.

“j” I didn’t see direct evidence presented that indicates that PKA activates Nf-kb signaling in this system. This would be important as several other cAMP/PKA inhibits NF-kb in some cell types. I also do not understand why PGE₂ seems to be coming from the lung in this diagram? Does this indicate that the predominant source is not the heart tissue itself?

Reply: We will correct the information on the lungs in the diagram. In fact, the lung is the major source of PGE₂ during envenomation, and we suppose that PGE₂ from cardiac tissue mainly acts locally. NF-kB activation was proposed based on previous results from Zoccal et al. 2016, but we have removed this information.

Figure 2

Is it odd that Propranolol had been previously reported to prevent selected aspects (contractile force coronary flow) of TsV y induced cardiac dysfunction? (see “EFFECTS OF TITYUS SERRULATUS SCORPION VENOM AND ONE OF ITS PURIFIED TOXINS (TOXIN y) ON THE ISOLATED GUINEA-PIG HEART” Silveira et al. 1991), and yet here it seems to not have had any effect on the parameters measured?

Reply: The use of propranolol, known as a beta-adrenoceptor antagonist, is suggested in the literature, as TsV induces the “catecholamine storm” with several studies that demonstrate its relevance, but mostly in rats. This relevance to rat models involves the fact that rat tonus is predominantly sympathetic; thus the autonomic axis is exacerbated after envenomation. However, in our mouse model, the use of propranolol is irrelevant, as shown in Figure 4. In clinics, propranolol is not used frequently, only when patients experience exacerbated cardiac disturbance related to high blood pressure. However, we found that the other division of the autonomic

system, namely the parasympathetic branch, is involved in cardiovascular dysfunction in envenomed mice, given that atropine administration improves the mortality index. As mentioned in the “Discussion” section of the new version of the manuscript, atropine has been used to manage cardiovascular manifestations induced by scorpion envenoming, but the mechanisms have not been described (Bawaskar et al, 1992). In the new version, given that the vagus nerve is the main parasympathetic nerve and an important source of acetylcholine in the heart, we investigated the effect of cervical vagotomy in envenomed mice and observed the blockage of parasympathetic signaling, just as with atropine, improved venom-induced cardiovascular dysfunctions and mortality (Figure 5g-h).

In “i” it seems as though the TsV vehicle groups are not included. Is there a reason for this?

Reply: We apologize, but we did not include this because the WT and *Il1r^{-/-}* mice that were inoculated with the vehicle did not exhibit any of the 5 signals that were used to describe the clinical score. However, as requested, we included this group in Fig.1a.

Does “L” suggest that Dex isn’t a great treatment, it seems that it would need to be given within 15 mins of being stung. Perhaps the therapeutic statements should be rewritten to downplay this given these data.

Reply: In fact, when DEX treatment is delayed, the immune response is already triggered by the venom and the subsequent mediators released (with the induction of ACh release). High doses of DEX should be given as soon as possible after envenomation, but serum treatment may still be necessary to neutralize toxins.

In “m”, is the point that the lung is the most important source of PGE₂?

Reply: Yes. In fact, the lung is the major source of PGE₂ during envenomation, and we suppose that PGE₂ from cardiac tissues mainly acts locally.

Based on the previous data presented, I’m surprised that PGE₂ administration didn’t increase Ach in the Il-1R^{-/-} mice. Does this not indicate that the model presented where PGE₂ acts via EP1/2 and (now ep4 as well based on the AH23848 data) increased ACh release is potentially not complete or incorrect? Is this a dose or time issue? Since mPGE₂s inhibition in “O” seems to reduce Ach release induced by TsV?

Reply: We appreciate these important questions raised by the referee. They will certainly make our manuscript more consistent scientifically. We apologize for the “Discussion” section; as commented before, the first version was transferred from a *Nature Medicine* publication (initially submitted as a “Letter”), and therefore, we could not discuss our data extensively. In the new version, we discussed it based on several neuro-immune circuits described recently by different groups and published in top journals (including *Nature Communications*). These pathways can be related to several pathological conditions, especially those that contain cardiovascular components. (Line 305). Moreover, in the new version, we performed new experiments based on the referee's questions and confirmed the results, which demonstrate that the muscarinic antagonist reverts the venom-induced cardiovascular dysfunctions. Instead of evaluating the neural tonus, we performed a surgical cervical vagotomy and notably observed that the surgical vagal blockage preserved the cardiovascular functions and clinical score after the venom inoculation (Fig 5). The new data confirm that parasympathetic signals are involved in cardiovascular dysfunction during the scorpion envenomation via cholinergic vagal signaling. Overall, our results show a new neuroimmune mechanism involved in the “neurotransmitter storm,” as described in clinical articles (Cupo et al., 2015; Isbister et al., 2014).

Minor points:

Use continuous line numbers.

Reply: Thank you for your suggestion. We have revised our manuscript as recommended.

In ext data2, I think the axis on “d” should read “% CD14+ cells of CD31- PDGFR+” and similar for TLR2 and TLR4.

Reply: We have revised our manuscript as recommended.

On page 4 ln 22 should read “with an elevated concentration...”

Reply: We have revised our manuscript as recommended.

Page 6 line 20 seems an odd choice of words “untreated-WT envenomed” should maybe be vehicle treated?

Reply: We have revised our manuscript as recommended.

Page 11 line 26 “The data express mean...” should be “The data are expressed as....”

Reply: We are thankful for your suggestions and observations. We have implemented the requested changes. Moreover, the English has been revised and improved.

Reviewers' Comments:

Reviewer #2:

Remarks to the Author:

The manuscript by Reis et al has been substantially revised to address the reviewer's comments with new vagotomy experiments and an expanded results and discussion section. These changes have improved the manuscript. However, this reviewer still has some unresolved issues:

1. In the discussion section (lines 237-240), the authors state that " both the parasympathetic/vagal and sympathetic systems are excessively activated after scorpion venom administration, only the blockage of the latter prevented cardiovascular alterations and mortality....". I think this should be the former as inhibition of the sympathetic system did not affect cardiovascular function.
2. In the discussion section (lines 245-247), the authors state that expression of inflammatory genes such as COX-1 and-2 are increased in response to the venom. Interestingly, the heatmap data also shows that the gene PTGER2 or the gene encoding the prostaglandin E2 EP2 receptor is upregulated after administration of TsV yet the authors completely ignore this data. Is only EP2 increased or do the authors also observe an upregulation of EP4? This should be added to the discussion.
3. In Figure 2, panels (a) and (b) for PGE2 and IL-1b are reversed according to the figure legend. Please revise either the legend or the figure. The same problem also exists for panels (g) and (h) in figure 2.
4. In Figure 3(c) the scale for ejection fraction is not consistent across all 3 panels which is misleading for the readership. Please make sure that they are all consistent.
5. In Figure 5(k) it is rather surprising that the amount of PGE2 in the lung is increased in the IL1r-/- mice relative to the C57 controls. How do the authors explain this finding? Additionally, throughout the manuscript, the authors imply that the PGE2 is coming from the lungs to affect cardiac function. Although the authors have previously reported the effect of the toxin on PGE2 levels in the lungs, what evidence is there to suggest that the PGE2 is coming from the lungs in this study? PGE2 is an autacoid that can be made by several cell types in the heart as well as infiltrating inflammatory cells. Thus, its concentration within the heart is likely to be higher than circulating concentrations (see fig 1 panels O and P). Therefore, this reviewer thinks that panel J in figure 2 is could be edited to show the possibility of it coming from other cell types or eliminated.
6. As a minor point, postganglionic is spelled incorrectly in figure 6.
7. As a minor point, the authors refer to mPGE2 synthase-1. This could be abbreviated to mPGES-1.

Reviewer #3:

Remarks to the Author:

This article represents the effort of an interdisciplinary group to investigate the role of the inflammatory response via the influence of the parasympathetic autonomic nervous system as a key component of cardiac dysfunction in scorpion envenomation.

They demonstrate the relationship between IL-1, PGE2 and acetylcholine mediators in the modulation of cardiac dysfunction and mortality and prove the preventive and additive action of dexamethasone in antiserum therapy to treat cardiac dysfunction.

The additive action of dexamethasone with the antiserum in the prevention of cardiac dysfunction represents a valid proof of concept to suggest its use in the control of the massive release of inflammatory mediators.

Reviewer #4:

Remarks to the Author:

The resubmitted manuscript "Interleukin-1 receptor-induced PGE2 production controls acetylcholine-mediated cardiac dysfunction and mortality during scorpion envenomation" by Reiss et al, although substantially improved remains problematic in a few critical areas.

First, I would like to acknowledge the significant amount of work in re-organizing the work and improving readability. This was certainly no easy task. Thank you.

There are always minor items that I may disagree with, but these are differences of opinion and do not prevent publication.

I think the only major issue is again that the heat maps are not appropriate. The raw data, that is presumably normalized data, shows that there is no significant change in several of the genes in fig 1m.

This includes Alox5ap which to my eyes I would have thought would be significant from the heat map. This should illustrate that heat maps are not able to demonstrate the variation and allow for easy comparison between groups. If the all of data is not significant so be it, but show it.

REVIEWERS' COMMENTS:

Reviewer #2 (Remarks to the Author):

The manuscript by Reis et al has been substantially revised to address the reviewer's comments with new vagotomy experiments and an expanded results and discussion section. These changes have improved the manuscript. However, this reviewer still has some unresolved issues:

1. In the discussion section (lines 237-240), the authors state that "both the parasympathetic/vagal and sympathetic systems are excessively activated after scorpion venom administration, only the blockage of the latter prevented cardiovascular alterations and mortality...". I think this should be the former as inhibition of the sympathetic system did not affect cardiovascular function.

Answer: We would like to thank the reviewer 2 for this important observation and we sorry for our mistake. We already corrected the sentence "...both the parasympathetic/vagal and sympathetic systems are excessively activated after scorpion venom administration, only the blockage of the first prevented cardiovascular alterations and mortality..." (in the new manuscript version: lines 250-252)

2. In the discussion section (lines 245-247), the authors state that expression of inflammatory genes such as COX-1 and -2 are increased in response to the venom. Interestingly, the heatmap data also shows that the gene PTGER2 or the gene encoding the prostaglandin E₂ EP2 receptor is upregulated after administration of TsV yet the authors completely ignore this data. Is only EP2 increased or do the authors also observe an upregulation of EP4? This should be added to the discussion.

Answer: We included the increase in the gene coding for EP2 receptor in the discussion. Unfortunately, we have not tested EP4 expression in these samples, however, the use of pharmacological antagonists for EP2 and EP4 indicate a role for both of them.

3. In Figure 2, panels (a) and (b) for PGE₂ and IL-1b are reversed according to the figure legend. Please revise either the legend or the figure. The same problem also exists for panels (g) and (h) in figure 2.

Answer: We apologize for that. We have now corrected this mistake in both figures.

4. In Figure 3(c) the scale for ejection fraction is not consistent across all 3 panels which is misleading for the readership. Please make sure that they are all consistent.

Answer: We thank you for this important observation. We standardize the scale bar for ejection fraction as requested.

5. In Figure 5(k) it is rather surprising that the amount of PGE₂ in the lung is increased in the IL1r^{-/-} mice relative to the C57 controls. How do the authors explain this finding? Additionally, throughout the manuscript, the authors imply that the PGE₂ is coming from the lungs to affect cardiac function. Although the authors have previously reported the effect of the toxin on PGE₂ levels in the lungs, what evidence is there to suggest that the PGE₂ is coming from the lungs in this study? PGE₂ is an autacoid that can be made by several cell types in the heart as well as infiltrating inflammatory cells. Thus, its concentration within the heart is likely to be higher than circulating concentrations (see fig 1 panels O and P). Therefore, this reviewer thinks that panel J in figure 2 is could be edited to show the possibility of it coming from other cell types or eliminated.

Answer: We thanks the suggestion and removed this panel.

6. As a minor point, postganglionic is spelled incorrectly in figure 6.

We are thankful for this important observation. We already corrected “posganglionic” for “postganglionic” in Figure 6.

7. As a minor point, the authors refer to mPGE₂ synthase-1. This could be abbreviated to mPGES-1.

Answer: We modified as suggested.

Reviewer #3 (Remarks to the Author):

This article represents the effort of an interdisciplinary group to investigate the role of the inflammatory response via the influence of the parasympathetic autonomic nervous system as a key component of cardiac dysfunction in scorpion envenomation.

They demonstrate the relationship between IL-1, PGE₂ and acetylcholine mediators in the modulation of cardiac dysfunction and mortality and prove the preventive and additive action of dexamethasone in antiserum therapy to treat cardiac dysfunction.

The additive action of dexamethasone with the antiserum in the prevention of cardiac dysfunction represents a valid proof of concept to suggest its use in the control of the massive release of inflammatory mediators.

Reviewer #4 (Remarks to the Author):

The resubmitted manuscript "Interleukin-1 receptor-induced PGE₂ production controls acetylcholine-mediated cardiac dysfunction and mortality during scorpion envenomation" by Reiss et al, although substantially improved remains problematic in a few critical areas.

First, I would like to acknowledge the significant amount of work in re-organizing the work and improving readability. This was certainly no easy task. Thank you.

There are always minor items that I may disagree with, but these are differences of opinion and do not prevent publication.

I think the only major issue is again that the heat maps are not appropriate. The raw data, that is presumably normalized data, shows that there is no significant change in several of the genes in fig 1m.

This includes Alox5ap which to my eyes I would have thought would be significant from the heat map. This should illustrate that heat maps are not able to demonstrate the variation and allow for easy comparison between groups. If the all of data is not significant so be it, but show it.

Answer: First, we would like to thank the referee for the kind words about our new manuscript version. As the referee suggested, we have removed the heat maps and included a bar figure showing only genes expression with statistically significant alterations (Figure 1). The heat map from Figure 2 are now only as a Supplementary Table.